



# Atmospheric chemistry in East Asia determines the iron solubility of aerosol particles supplied to the North Pacific Ocean

Kohei Sakata[1,a*], Shotaro Takano[2], Atsushi Matsuki[3], Yasuo Takeichi[4], Hiroshi Tanimoto[1,5], Aya Sakaguchi[6], Minako Kurisu[7,b], and Yoshio Takahashi[8,9]

[1]Earth System Division, National Institute for Environmental Studies, 16-2 Onogawa, Tsukuba, Ibaraki 305-8506, Japan.

[2]Institute for Chemical Research, Kyoto University, Kyoto 611-0011, Japan.

[3]Institute of Nature and Environmental Technology, Kanazawa University, Kakuma, Kanazawa, Ishikawa 920-1192, Japan.

[4]Department of Applied Physics, Graduate School of Engineering, Osaka University, 2-1 Yamadaoka, Suita, Osaka 565-0871, Japan.

[5]Graduate School of Environmental Studies, Nagoya University, Furo-cho, Chikusa-ku, Nagoya 464-8601, Japan

[6]Institute of Pure and Applied Sciences, University of Tsukuba, 1-1-1 Tennodai, Tsukuba, Ibaraki 305-8577, Japan.

[7]Submarine Resources Research Center, Japan Agency for Marine-Earth Science and Technology (JAMSTEC), 2-15, Natsushima-cho, Yokosuka, Kanagawa, 237-0061, Japan.

[8]Graduate School of Science, The University of Tokyo, 7-3-1, Hongo, Bunkyo-ku, Tokyo 113-0033, Japan.

[9]Institute of Materials Structure Science, High-Energy Accelerator Research Organization Tsukuba, Ibaraki 305-0801, Japan.

[a]now at: Institute of Pure and Applied Sciences, University of Tsukuba, 1-1-1 Tennodai, Tsukuba, Ibaraki 305-8577, Japan.

[b]now at: Atmosphere and Ocean Research Institute, The University of Tokyo, 5-1-5, Kashiwanoha, Kashiwa, Chiba 277-8564, Japan.

*Corresponding author: Kohei Sakata.

**Email:** kohei.sakata.33@outlook.jp

ORCID: 0000-0002-0103-9631





**Abstract.**

The deposition of dissolved iron (d-Fe) from East Asian aerosols to the North Pacific Ocean modulates primary productivity in surface waters, facilitating uptake of atmospheric carbon dioxide by the ocean, thereby impacting global climate. Since the microorganisms in the surface seawater utilize d-Fe as a micronutrient, bioavailability of aerosol Fe depends on its solubility

30 ($Fe_{sol}\%$). Although $Fe_{sol}\%$ is influenced by both emission sources and atmospheric processing, their effects on $Fe_{sol}\%$ are not fully understood. We assessed the factors controlling $Fe_{sol}\%$ in size-fractionated aerosol particles collected along the Sea of Japan coast for one year (July 2019–June 2020). Approximately 70% of d-Fe in East Asian aerosols was present in fine aerosol particles (<1.3 µm), with $Fe_{sol}\%$ ranging from 4.1% to 94.9%. Anthro-Fe accounted for about 50% of d-Fe in fine aerosol particles during periods outside the COVID-19 lockdown, but its contribution was negligible during the lockdown. The $Fe_{sol}\%$

35 in fine aerosol particles correlated with the abundance of water-soluble Fe species (Fe(II, III)-sulfates and Fe(III)-oxalate). These water-soluble Fe species were detected in both mineral dust and anthropogenic aerosols in fine aerosol particles. Dissolution models optimized for Fe in mineral dust and anthropogenic aerosols showed that Fe in both aerosol types dissolved by proton-promoted dissolution under acidic conditions (pH < 2.0). Subsequently d-Fe dissolved from aerosols was stabilized by the formation of Fe(III)-oxalate in the aqueous phase. Thus, comprehensive understandings of the chemical alteration

40 processes of East Asian aerosols are essential for accurately quantifying their $Fe_{sol}\%$ upon transport to the North Pacific.



## 1. Introduction

Primary production in high nutrient-low chlorophyll (HNLC) regions is limited by the depletion of dissolved iron (d-Fe, Martin et al., 1994; Jickells et al., 2005; Boyd et al., 2007). Ocean iron (Fe) fertilization can modulate primary production in the euphotic zone, thereby increasing the uptake of carbon dioxide and potentially exerting a significant influence on the global climate system. (Martin, 1990; Martin et al., 1994; Falkowski et al., 2000; Jickells et al., 2005; Boyd et al., 2007). Atmospheric deposition of aerosol Fe is a dominant source of d-Fe in the North Pacific surface seawater. Given that microorganisms in surface seawater utilize d-Fe as a nutrient (Moore et al., 2013), the bioavailability of aerosol Fe is highly dependent on fractional Fe solubility ($Fe_{sol}\% = (d\text{-}Fe/total\ Fe) \times 100$) (Sholkovitz et al., 2012; Mahowald et al., 2018). The values of $Fe_{sol}\%$ in aerosol particles vary considerably (0.1–90%), but the factors controlling the $Fe_{sol}\%$ have not been fully understood (Sholkovitz et al., 2012; Mahowald et al., 2018).

One of the factors controlling $Fe_{sol}\%$ in aerosol particles is the difference in $Fe_{sol}\%$ between Fe in mineral dust and anthropogenic aerosols emitted through high-temperature combustion (Sholkovitz et al., 2012; Mahowald et al., 2018; Ito et al., 2021). Although the annual emission of anthropogenic Fe (anthro-Fe) was about an order of magnitude smaller than that of Fe in mineral dust (mineral-Fe), anthro-Fe is a possible source of d-Fe in surface water because it exhibits a higher $Fe_{sol}\%$ (up to 80%) than mineral-Fe ($Fe_{sol}\% < 1\%$; Myriokefalitakis et al., 2018; Hamilton et al., 2019; Ito et al., 2021). Indeed, the high $Fe_{sol}\%$ associated with anthro-Fe has been observed from East Asia aerosols, especially in fine aerosol particles (Kurisu et al., 2016, 2021, 2024; L. Liu et al., 2022; Hsieh et al., 2022; Sakata et al., 2023). However, the contribution of anthro-Fe to d-Fe in aerosol particles has not been well evaluated quantitatively through field observations. Furthermore, mineral-Fe and anthro-Fe undergo atmospheric processes, including proton-promoted, ligand-promoted, and photoreductive dissolutions, during atmospheric transport, which elevate their $Fe_{sol}\%$ (Journet et al., 2008; Shi et al., 2011a, 2015; Paris et al., 2011; Chen and Grassian, 2013; Ito and Shi, 2016; Li et al., 2017; Sakata et al., 2022). Single-particle analyses have shown that mineral-Fe and anthro-Fe in fine aerosol particles are internally mixed with sulfate, nitrate, and organic matter, including oxalate (Li et al., 2017; Sakata et al., 2022; Zhang et al., 2019; Zhou et al., 2020; Y. Zhu et al., 2020, 2022; Xu et al., 2023; Ueda et al., 2023). These internally mixed particles provide evidence of the chemical alteration of Fe-containing aerosols in the atmosphere, but determining the $Fe_{sol}\%$ through single-particle analysis remains a challenging task (Ueda et al., 2023). Therefore, net effect of atmospheric processes of Fe-containing particles mixed with acidic species and organic matters on $Fe_{sol}\%$ remains poorly understood.

It is well-known that East Asia is one of the world's largest sources of mineral-Fe and anthro-Fe transported to the North Pacific Ocean, which is one of the HNLC regions (Myriokefalitakis et al., 2018; Hamilton et al., 2019; Ito et al., 2021). Additionally, East Asia continues to grapple with air pollution problems, and it has been reported that anthropogenic $SO_2$ and other pollutants are causing chemical alteration of mineral-Fe and anthro-Fe in the atmosphere over urban areas in East Asia (Li et al., 2017: Zhu et al., 2020, 2022; Xu et al., 2023). Given that the Fesol% in aerosol particles supplied to the North Pacific is mainly controlled by processes occurring during transport between East Asia and Japan (Buck et al., 2013; Sakata et al., 2022), long-term observations in Japan, located at the eastern edge of East Asia, are crucial for providing insights into these





controlling factors. Therefore, this study monthly collected seven size-fractionated aerosol particles at the Noto Ground-based Research Observatory (NOTOGRO) along the Sea of Japan coast from July 2019 to June 2020. NOTOGRO is a suitable location for collecting long-range-transported aerosols minimally influenced by local emissions (Fig. 1, Sakata et al., 2021). The sampling period encompassed the COVID-19 lockdown period in China when anthro-Fe concentration in China was

considerably decreased (from the end of January to February 2020; Liu et al., 2021; Li et al., 2021; Zheng et al., 2020; Xu et al., 2022). Considerable anthro-Fe concentration reduction due to this contingency provided us opportunities to assess the effect of anthropogenic activities on $Fe_{sol}$% in aerosol particles in East Asia. Using these samples, this study conducted various analyses related to estimating Fe sources and alteration processes to understand factor controlling $Fe_{sol}$% in East Asia region. Atmospheric concentrations of total and dissolved metals were determined using high-resolution inductively coupled plasma

mass spectrometry (HR-ICP-MS). The contributions of mineral-Fe and anthro-Fe to d-Fe were estimated by (i) positive matrix factorization (PMF, Norris et al., 2014) and (ii) the molar ratio of d-Fe relative to dissolved Al ([d-Fe]/[d-Al]) as a new indicator for sources and dissolution processes of d-Fe in aerosol particles (Fig. S1. Sakata et al., 2023). To identify d-Fe species in aerosol particles, representative Fe species were determined by macroscopic X-ray absorption near-edge structure (XANES) spectroscopy and then relationship between Fesol% and Fe species were investigated. Furthermore, spot analyses of Fe species

in mineral dust and anthropogenic aerosols were performed by microfocused XANES combined with X-ray fluorescence mapping (μ-XRF-XANES) to assess the alteration processes of Fe. Finally, dissolution kinetic models optimized for Fe in mineral dust and anthropogenic aerosols were used to the effect of pH on the Fe dissolution from these aerosols. From these results, the influence of the Fe source (mineral-Fe or anthro-Fe) on $Fe_{sol}$% in aerosol particles transported to the North Pacific was evaluated independently.


## 2.    Material and Methods

### 2.1.    Aerosol sampling.

NOTOGRO is located in the coastal region of the Sea of Japan in Suzu City, Japan (37.4513°N, 137.3589°E; Fig. 1). The city lacks industrial or other anthropogenic emission sources. Size-fractionated aerosol samples were collected using a high-

volume air sampler (Model-120, Kimoto, Japan) equipped with a Sierra-type cascade impactor (TE-236, Tisch Environmental Inc., USA). The air sampler was installed on the rooftop 10 m above ground level. Aerosol particles were collected separately in seven stages (>10.2, 4.2–10.2, 2.1–4.2, 1.3–2.1, 0.69–1.3, 0.39–0.69, and <0.39 μm) with a flow rate of 0.566 m³ min⁻¹. Custom-made polytetrafluoroethylene filters were used as the sampling filter for all stages (PTFE, PF050, Advantech, Japan, Sakata et al., 2018, 2021). The PTFE filters were hydrophilized with ethanol (99.5%, Wako First Class, Wako, Japan).

Subsequently, the hydrophilized PTFE filters were soaked in 1 mol/L hydrochloric acid (EL grade, Kanto Chemical) and heated at 180°C for one day. After that, the filters were placed in ultrapure water and heated at 180°C for one day. The rinsed filters were then air-dried in a clean booth. The hydrophobicity of the PTFE filters restored by air-drying due to the complete removal of ethanol from the filters. The rinsed and dried PTFE filters were stored in polyethylene bags. The blank Fe





concentration in PTFE filter was 0.438±0.713 ng cm$^{-2}$ for acid digestion and 0.044±0.040 ng cm$^{2-}$ for ultrapure water

extraction. These blank concentrations were at least an order of magnitude lower than the blank Fe concentration in cellulose

filters (Morton et al., 2013). The filter blanks for Fe at the average sampling flow in this study (approximately 5000 m$^3$) was

less than 0.1 pg m$^{-3}$ and had little effect on the Fe concentration in the aerosol samples.

   Aerosol samples were collected monthly from July 2019 to June 2020 (Table S1). The filters with aerosol samples were

folded in half, placed in polyethylene bags, and then stored in a desiccator (RH < 20%). Since the COVID-19 lockdown in

China run from January 23 to February 19 (Liu et al., 2021; Li et al., 2021; Zheng et al., 2020; Xu et al., 2022), the aerosol

samples in January and February were collected during and after the lockdown (Table S1). The status of COVID-19 lockdown

for other samples is shown in Table S1.

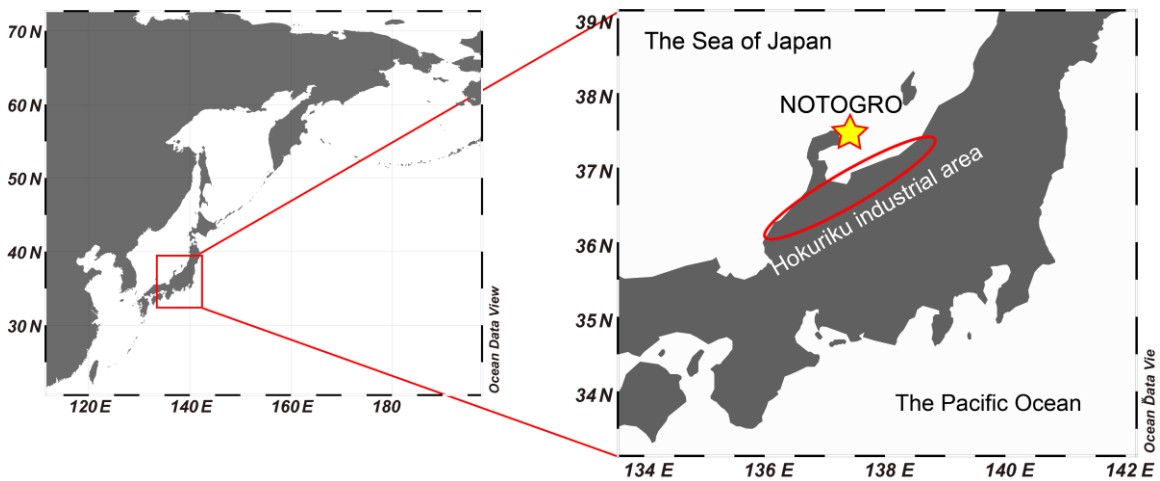

**Figure 1.** The sampling site (NOTOGRO) of size-fractionated aerosol sampling.

### 2.1.1.    Determinations of total and dissolved metal concentrations

   In this study, the Fe concentration measured after complete acid digestion of aerosol particles is denoted as T-Fe (= d-Fe +

insoluble Fe). About one-fifth of the collected aerosol sample in each stage was decomposed by a mixed acid (2 mL of 15.5

mol L$^{-1}$ HNO$_3$, 2 mL of 11.3 mol L$^{-1}$ HCl, and 1 mL of 28 mol L$^{-1}$ HF, ultrapure AA 100, Kanto Chemical, Co., Inc., Japan)

in perfluoroalkoxy alkane (PFA) vials by heating at 150 °C for one day. The mixed acid was evaporated to dryness, and then

the evaporated residue was redissolved in 2% of HNO$_3$. The solutions were filtrated by a hydrophilic polyethersulfone filter

(PES filter; Millex, $\phi$: 0.45 µm, Merck, Germany). D-Fe in aerosol particles was extracted by 2–4 mL of ultrapure water in

polypropylene centrifuge tubing with horizontal shaking for one day. The extracted solution was then filtered through the PES

filter. After the evaporation to dryness, the evaporated residue was then redissolved in 2% HNO$_3$. Elemental concentrations

were determined using an HR-ICP-MS (Elemental II, Germany). The precision and accuracy of quantifications of target





elements were confirmed by repetitive analysis of reference material of urban aerosol (Table S2, NIES CRM 28. Urban aerosol, Mori et al., 2008). All sample treatments described above were performed in a clean room (class 1000).

To evaluate emission sources of Fe, the enrichment factor of Fe (EF$_{T-Fe}$) normalized by the Fe/Al ratio of upper continental crust (UCC) was calculated by the following equation:

$$EF_{T-Fe} = \frac{(T-Fe/T-Al)_{aerosol}}{(Fe/Al)_{UCC}} \quad \text{(eq. 1)}$$

In this study, considering the variation of Fe/Al in the UCC, the average value from five literature sources (=0.52± 0.12) was used (Turkian and Wedepohl, 1961; Taylor, 1964; Wedepohl, 1995; Taylor and McLennan, 1995; Rudnick and Gao, 2003). The contribution of anthropogenic Fe has traditionally been considered significant when EF$_{T-Fe}$ exceeds 10. However, recent studies have indicated a narrow range of Fe/Al ratios in Asian dust (X. Liu et al., 2022; Sakata et al., 2023). Consequently, this study adopts a more conservative threshold, recognizing anthro-Fe contributions when EF$_{T-Fe}$ is greater than 2.0 (Fe/Al > 1.04; Sakata et al., 2023). In addition to EF$_{T-Fe}$, T-Fe concentrations associated with mineral dust and anthro-Fe were estimated by the following equations:

$$Mineral\ Fe = Aerosol\ Al \times (Fe/Al)_{crustl} \quad \text{(eq. 2)}$$

$$Anthropogenic\ Fe = Aerosol\ Fe - Mineral\ Fe \quad \text{(eq. 3)}$$

### 2.2. The source apportionment of T-Fe and d-Fe by a diagram between EF of Fe and [d-Fe]/[d-Al]

The diagrams of [d-Fe]/[d-Al] combined with EF$_{T-Fe}$ is a useful tool for evaluating the sources and dissolution processes of d-Fe in aerosol particles because Fe$_{sol}$% values of the aerosol particles vary depending on the dominant sources of T-Fe and d-Fe (Sakata et al., 2023). The T-Fe and d-Fe sources can be categorized into the following five groups (Fig. 2). In the first group, T-Fe is primarily associated with mineral dust (EF$_{T-Fe}$ < 2.0). Under conditions of proton-promoted dissolution, the [d-Fe]/[d-Al] ratios of aluminosilicate minerals (*e.g.*, biotite, illite, and chlorite) were ranged from 0.14 to1.03 (Kodama and Schnitzer, 1973; Lowson et al., 2005; Bibi et al., 2011; Bray et al., 2015). Furthermore, the [d-Fe]/[d-Al] ratio of Asian dust was 0.24 ± 0.20 (Duvall et al., 2008). From these reported values, the range of [d-Fe]/[d-Al] ratio of proton-promoted dissolution of mineral dust was defined from 0.10 to 1.00 (brown area in Fig. 2). The T-Fe in the second group is also derived from mineral dust, but the d-Fe in this group is mainly dissolved by ligand-promoted dissolution. The [d-Fe]/[d-Al] ratio in the group exceeds 1.00 owing to the preferential complexation of iron by organic ligands over Al (green area in Fig. 2, Kodama and Schnitzer, 1973; Bray et al., 2015).

Third group represent a binary mixing of mineral dust and insoluble anthro-Fe, which is characterized by EF$_{T-Fe}$ > 2.0 and a [d-Fe]/[d-Al] ratio <1.00 (white area in Fig. 2). Here, anthro-Fe refers to anthropogenic Fe-rich particles that can increase EF$_{T-Fe}$, including Fe-oxide nanoparticles, which emits from not only high-temperature combustion processes (*e.g.*, steel industry, coal combustion; Ito et al., 2021) but also non-combustion sources such as debris from automobile brake pads (Li et al., 2022; Fu et al., 2023). The anthro-Fe present in this group exhibits low solubility and thus makes a negligible contribution to the increase in the [d-Fe]/[d-Al] ratio observed in the aerosols. Therefore, it is inferred that the d-Fe primarily reflects the




values characteristic of mineral particles with which the insoluble anthro-Fe is associated. Unlike third group, the [d-Fe]/[d-Al] ratio in the fourth group is greater than 1.0 because of the high $Fe_{sol}$% of anthro-Fe. As a result, the fourth group is characterized by aerosols where both T-Fe and d-Fe are influenced by anthro-Fe. (grey area in Fig. 2). Finally, final group is aluminosilicate glasses emitted from high-temperature combustions, including coal combustions. It is known that aluminosilicate glasses were emitted from high-temperature combustions, which can be characterized low $EF_{T-Fe}$ (< 2.0) and

[d-Fe]/[d-Al] ratio (<0.10), which are totally different those for anthropogenic Fe-rich particles. Thus, one of the key advantages of this method lies in its capacity to discriminate between anthropogenic Fe-rich particles and aluminosilicate glasses produced by high-temperature combustion processes.

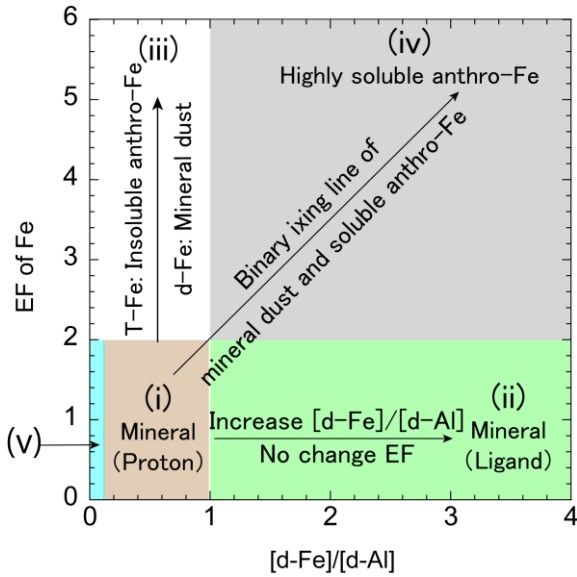

**Figure 2.** The diagram of $EF_{T-Fe}$ and [d-Fe]/[d-Al] ratio for evaluating T-Fe and d-Fe sources of aerosol particles.


As detailed in Section 3.3.2, the d-Fe in fine aerosol particles was composed of a binary mixing of d-Fe dissolved by proton-promoted dissolution of mineral dust and highly soluble anthro-Fe. Here, the contribution of anthro-Fe to d-Fe ($F_{anthro}$) in fine aerosol particles was estimated on the basis of a two-component mixing model using the following equations (Sakata et al., 2023):

$$F_{mineral} + F_{anthro} = 1, \text{(Eq. 4)}$$

$$\left(\frac{[d-Fe]}{[d-Al]}\right)_{aerosol} = \left(\frac{[d-Fe]}{[d-Al]}\right)_{mineral} \times F_{mineral} + \left(\frac{[d-Fe]}{[d-Al]}\right)_{anthro} \times F_{anthro} \quad \text{(eq. 5)}$$

The average [d-Fe]/[d-Al] ratio of coarse aerosol particles (=0.28) was the representative value of the $([d-Fe]/[d-Al])_{mineral}$. The representative [d-Fe]/[d-Al] ratio of anthro-Fe was 2.18, which was the average [d-Fe]/[d-Al] ratio of fine aerosol particles with a value higher than 1.50 (Sakata et al., 2023). Then, concentrations of d-Fe associated with mineral dust and anthro-Fe

were calculated by the following equations:





$$\text{Mineral } d-Fe = d-Fe \times F_{mineral} \qquad \text{(eq. 6)}$$

$$\text{Anthropogenic } d-Fe = d-Fe \times F_{anthro} \qquad \text{(eq. 7)}$$

Subsequently, the $Fe_{sol}\%$ of mineral dust (mineral-$Fe_{sol}\%$) was calculated by dividing mineral d-Fe by the mineral Fe, using the same approach as for anthropogenic Fe (anthro-$Fe_{sol}\%$).


### 2.3. Positive matrix factorization

Source apportionment of T-Fe and d-Fe in fine aerosol particles were performed by positive matrix factorization (EPA PMF version 5.0 Norris et al., 2014). The PMF analyses were performed separately for the entire sampling period (JPN+EAout), JPN, and EAout periods. The PMF analysis for the JPN+EAout was conducted to evaluate the monthly trend

of the normalized contribution of each factor, with the average of all contributions for each factor normalized to 1. In contrast, PMF analyses were performed separately for the JPN and EAout periods to evaluate the average $EF_{T\text{-}Fe}$, $Fe_{sol}\%$, and [d-Fe]/[d-Al] ratios of each factor. These analyses used fine aerosol particles collected during the respective periods. Input data for PMF analysis are concentrations of Na, Mg, Al, d-Al, K, Ca, Ti, V, Cr, Mn, Fe, anthro-Fe, d-Fe, Co, Ni, Cu, Zn, Sr, Cd, Ba, Pb, and $SO_4^{2-}$ in the fraction of <0.39, 0.39–0.69, and 0.69–1.3 μm). The input file for PMF analysis was concentrations of target

species and their uncertainties. Uncertainties of each element were evaluated by the following equations:

$$Uncertainty = \frac{5}{6} \times MDL \qquad \text{(eq. 8)}$$

$$Uncertainty = \sqrt{(Error\ fraction \times concentraion)^2 + (0.5 \times MDL)^2} \ \text{(eq. 9)}$$

where MDL is the method detection limit, defined as three times the standard deviation of the filter blank concentration. Equations 8 and 9 were used when target species concentrations were lower and higher than MDL, respectively. The PMF

analysis allows for three categories: "Strong", "Weak", and "Bad". These categories were typically chosen based on the signal-to-noise (S/N) ratio. The "Weak" category is selected when the S/N is between 0.5 and 1.0, and the "Bad" category is used if the S/N ratio is lower than 0.5. Species classified as "Weak" had their associated uncertainties tripled, and species classified as "Bad" were excluded from further analysis. Initially, PMF analysis was performed with all elemental categories set to "Strong" because the S/N for all species was higher than 7.0. Consequently, the coefficient of determination ($r^2$) between the

observed and modeled concentrations of the input species was greater than 0.60, with the exception of Cr in the EAout period (Tables S3 and S4). The PMF analysis for the EAout period was then rerun with the Cr category set to "Weak," but the results did not change significantly. Therefore, this study employed the PMF results with all species categories set to "Strong," based on the conventional use of the S/N ratio for category determination.

### 2.4. Macroscopic and micro-focused XAFS.

The Fe K-edge XANES spectra of the aerosol samples (7050–7300 eV) were recorded at BL-9A and BL-12C, Photon Factory (PF), High Energy Acceleration Research Organization (KEK), Ibaraki, Japan. The synchrotron radiation generated by the bending magnet was monochromatized by a double-crystal monochromator of Si(111). The XANES experiment were



performed in ambient air at room temperature. Approximately one-tenth of collected aerosol particles on PTFE filters were

transferred to a double-face carbon tape. The aerosol samples were oriented at 45° to the incident X-ray beam. The incident X-ray energy was calibrated with the peak top of the pre-edge peak of the nonderivative Fe K-edge XANES spectrum for hematite aligned to 7112 eV. All XANES spectra of aerosol samples were recorded in fluorescence yield mode. Fluorescence X-ray from the aerosol sample was detected with a seven-element silicon drift detector equipped with a Soller slit to reduce elastic X-ray around the beam pass. The front face of the Soller slit was covered with a 0.2 mm-thick PTFE filter to remove

fluorescence X-rays of coexisted elements (*e.g.*, Ca and Mn) and argon in the ambient air. Linear combination fitting of the XANES spectra of aerosol samples using reference materials was performed with REX2000 software. The fitting was performed over the energy range of 7100–7200 eV. The goodness of fit was evaluated by the following equation:

$$\Sigma R \; = \; \frac{\Sigma [I_{obs}(E) - I_{cal}(E)]^2}{\Sigma [I_{obs}(E)]^2} \qquad \text{(eq. 10)}$$

where $I_{obs}(E)$ and $I_{cal}(E)$ are X-ray absorption of the normalized X-ray absorptions of the samples and the calculated values at

each energy.

The μ-XRF-XANES analyses were performed at BL-15A1 in PF. Aerosol samples with sizes of 0.39–0.69 μm and 2.1–4.2 μm, collected in September 2019, were used for the μ-XRF-XANES analyses. The beam size at the sampling position (20×20 μm$^2$) is larger than the aerodynamic diameter of the target samples. Although these experiments were not single-particle analyses, spot analysis combined with XRF mapping allows for the identification of chemical species of target

elements from different emission sources (*e.g.*, mineral and non-mineral materials). Aerosol particles on the carbon tape were mounted on an acrylic sample holder and oriented at 45° to the direction of the incident X-ray beam. XRF maps of the 3d transition metals (Mn, Fe, Ni, Cu, and Zn) and light elements (Ti, Ca, K, Cl, and S) were acquired using a raster scan of the sample stage irradiated with 14 and 5.1 keV incident X-rays, respectively. Measurement spots for Fe species were selected based on the XRF maps of the target elements normalized by the incident X-ray intensity. Iron K-edge XANES spectra of the

regions of interest were recorded in quick-scan mode with a scan time of 180 sec. The same spectral analysis procedure used for macroscopic XANES was applied to the micro-focused XANES data.

## 2.5. Estimation of dissolution pH of mineral dust and anthro-Fe

### 2.5.1. Dissolution pH for mineral dust

Aerosol particles are repeatedly incorporated into and re-emitted from cloud water in the atmosphere (aerosol-cloud cycles), with Fe dissolution primarily occurring in highly acidic aerosol phases (Spokes et al., 1994; Shi et al., 2015; Maters et al., 2016). Given that the dissolution of Fe in mineral dust occurs in the aerosol phase (pH < 3.0), Fe dissolution from mineral dust was simulated using the three Fe-pools model (Shi et al., 2011a; Sakata et al., 2023). The fast Fe pool (ferrihydrite and poorly crystalline Fe oxides), the intermediate Fe pool (Fe oxide nanoparticles), and the slow Fe pool

(crystalline Fe oxides and aluminosilicates) represent three Fe pools with different dissolution rates (*k*, Table 1). Shi et al. (2011) reported that the dissolution rate of the slow Fe-pool is similar to that of illite. However, in aerosol samples





collected for this study and our previous work (Sakata et al., 2022), biotite is more abundant than illite. Given that the

dissolution rate of biotite is approximately one order of magnitude higher than that of illite (Bibi et al., 2011; Bray et al.,

2015), we set the dissolution rate of the slow Fe-pool one order of magnitude higher than in the original model. Therefore,

in this study, the dissolution rate of the slow Fe-pool is set one order of magnitude higher than that of the original model.

Assuming the first-order reaction, the molar concentration at time ($t$) ($[\text{d-Fe}(t)]$) is described in the following equation:

$$[\text{d} - \text{Fe}]_{\text{mineral}}\ (\mu mol\ g^{-1}) = [\text{d} - \text{Fe}]_{\text{fast}} + [\text{d} - \text{Fe}]_{\text{intermediate}} + [\text{d} - \text{Fe}]_{\text{slow}} \quad \text{(eq. 11)}$$

$$[\text{d} - \text{Fe}(t)]_{\text{fast}}\ (\mu mol\ g^{-1}) = [\text{d} - \text{Fe}]_{\text{mineral}} \times [\%\text{FeT}]_{\text{fast}} \times (1 - e^{-kt}) \quad \text{(eq. 12)}$$

where $[\text{d-Fe}]_{\text{mineral}}$ refers to d-Fe concentration in mineral dust calculated using eq. 6, and the unit conversion from ng

$m^{-3}$ to $\mu mol\ g^{-1}$ for d-Fe concentration are provided in the S1.1 in Supplementary Information. The [%FeT] denotes the

maximum percentage of Fe that can be solubilized, and $k$ represents the rate constant ($h^{-1}$). The pH dependence of these

parameters is presented in Table 1. The reaction time, $t$, was set to 54 hours, taking into account atmospheric transport

and the aerosol-cloud cycles (Sakata et al., 2023). Finally, the pH value for which the sum of d-Fe concentrations across

all pools equalled $[\text{d-Fe}]_{\text{aerosol}}$ was determined.


**Table 1 pH dependence of parameters for three Fe-pool model**

| Fe pool | %FeT (%) | Dissolution rate ($h^{-1}$) |
|---|---|---|
| Fast | pH 1.0-2.0: Fixed at 0.9%<br>pH 2.0-3.0: %FeT $= -0.4 \times$ pH $+1.7$ | log $k_{\text{fast}} = -0.50 \times$ pH $+ 1.87$ |
| Intermediate | pH 1.0-2.0: Fixed at 3.0%<br>pH 2.0-3.0: %FeT $= -2.0 \times$ pH $+ 7.0$ | log $k_{\text{intermediate}} = -0.66 \times$ pH $+ 0.36$ |
| Slow | pH 1.0-3.0: %FeT $= -15.2 \times$ pH $+ 58.4$ | log $k_{\text{slow}} = -0.44 \times$ pH $- 0.76$ |

### 2.5.2. Dissolution pH for anthro-Fe

Using hematite nanoparticles as a proxy for anthro-Fe, the dissolution pH of anthro-Fe was estimated under the

assumption that anthro-Fe dissolution was solely driven by proton-promoted dissolutions. Based on the assumption that the

S/L ratio for anthro-Fe is 0.06 g/L, which was comparable to that of mineral dust, the aerosol liquid water (ALW) content

associated with hematite nanoparticles was quantified using the following equation.

$$ALW\ (L\ m^{-3}) = \frac{Anthro-Fe\ concentration/0.699}{0.06\ (=\frac{S}{L}ratio)} \quad \text{(eq. 13)}$$

where 0.699 is the mass fraction of Fe in hematite nanoparticles, and anthro-Fe concentrations were estimated by eq. 3. The

pH dependence of the anthro-$\text{Fe}_{\text{sol}}$% in ALW under the equilibrium state was estimated based on the solubility product of

hematite nanoparticle (Bonneville et al., 2004). The proton-promoted dissolution of hematite nanoparticles and the solubility

product ($^{*}K_{\text{so}} = 0.52$) of this reaction are described as the followings:





$$\frac{1}{2}\text{Fe}_2\text{O}_3 + 3\text{H}^+_{(aq)} \leftrightarrow \text{Fe}^{3+}_{(aq)} + \frac{3}{2}\text{H}_2\text{O} \ (\text{eq. 14})$$

$$log \ ^*K_{SO} = log[a_{Fe^{3+}}] + npH \ (\text{eq. 15})$$

where n is reaction order determined by previous study (n: 2.85, Bonneville et al., 2004). The $[a_{Fe^{3+}}]$ represents the activity

of $Fe^{3+}$ in the ALW. To simplify the calculations, the activity coefficient is 1, which means that the $[a_{Fe^{3+}}]$ is considered to be equal to the Fe concentration in the solution (nmol $L^{-1}$). The $[a_{Fe^{3+}}]$ in ALW at each pH was calculated by substituting pH values into Equation 15. Subsequently, the anthro-$Fe_{sol}$% at equilibrium was calculated for various pH values using the following equation.

$$Equilibrium \ anthro - Fe_{sol}\% = \frac{ALW \ (L \ m^{-3}) \times [a_{Fe^{3+}}] \ (nmol \ L^{-1})}{anthropogenic \ Fe \ (ng \ m^{-3})} \times 100 \ (\text{eq. 16})$$

The pH at which the equilibrium anthro-$Fe_{sol}$% matched the actual anthro-$Fe_{sol}$% was determined and defined as the pH exhibited during the leaching of anthro-Fe.

## 3. Results and Discussion

### 3.1. Backward and forward trajectories

This study categorized aerosol samples into two groups based on backward and forward trajectory analyses (Figs. S2 and S3). The first group was defined as the Japanese air mass (JPN period: July–October 2019 and May–June 2020). Air masses arriving at the sampling site (NOTOGRO) during the JPN period originated from the domestic region of Japan and its marginal sea (Fig. S2a). In addition, forward trajectory analyses indicated that these air masses were not transported to the North Pacific Ocean (Fig. S3a). The second group was defined as the East Asian outflow period (EAout period: November 2019–April 2020;

seasons: winter and spring). During the EAout period, air masses arriving at the sampling site originated from East Asia and were subsequently transported to the Pacific Ocean (Figs. S2b and S3b).

### 3.2. Monthly variation and size distributions of T-Fe and EF$_{T-Fe}$

    Atmospheric T-Fe concentration in total suspended particulates (TSPs: sum of the all-size fractions) ranged from 15.6 to

312 ng m$^{-3}$ (Fig. 3a, average ± standard deviation (avg ± 1σ: 113 ± 108 ng m$^{-3}$). Coarse aerosol particles (>1.3 μm) accounted for 84.8 ± 5.6% of T-Fe in TSPs (Fig. 3a). Concentrations of T-Fe and typical mineral elements (*i.e.*, Al, Ti, and non-sea-salt Ca$^{2+}$) were higher from February to April than in other seasons due to long-range transportation of Asian dust (Figs. 3a and S4, Uematsu et al., 1983; Zhu et al., 2020; Kawai et al., 2021). The annual average of EF$_{T-Fe}$ in TSP samples was 1.3 ± 0.3 (Fig. 3a), which was identical to that for coarse aerosol particles (EF$_{T-Fe}$: 1.3 ± 0.4, Fig. 2b). This result indicated that T-Fe in

TSP and coarse aerosol particles were mainly derived from mineral dust.

    T-Fe concentrations in fine aerosol particles (sum of <0.39 μm, 0.39–0.69 μm, and 0.60–1.3 μm fractions) varied from 4.1 ng m$^{-3}$ to 31.7 ng m$^{-3}$ (avg ± 1σ: 14.0 ± 10.2 ng m$^{-3}$, Fig. 3a). The annual average of EF$_{T-Fe}$ was 2.2 ± 1.0, indicating that anthro-Fe was one of the sources of T-Fe in fine aerosol particles (Fig. 3b). The relative contribution of anthro-Fe to T-Fe in





fine aerosol particles was the largest in the 0.39–0.69 μm size fraction, owing to the high $EF_{T\text{-}Fe}$ (Fig. 3b). This result is

consistent with previous studies using the Fe isotope ratio (Kurisu et al., 2016). The $EF_{T\text{-}Fe}$ in the fine aerosol particles showed distinct seasonal variations, with higher values during the JPN period (2.9 ± 0.8) than during the EAout period (1.5 ± 0.5; Fig. 3b). This result indicated that relative abundance of anthro-Fe to T-Fe in fine aerosol particles was higher during the JPN period than during the EAout period, likely due to the greater contribution of Asian dust in spring (Fig. 3a). However, this lower relative abundance of anthro-Fe to T-Fe did not necessarily indicate a lower absolute concentration of anthro-Fe. Indeed,

the absolute anthro-Fe concentration in fine aerosol particles during the JPN period (avg ± 1σ: 6.6 ± 5.6 ng m$^{-3}$, range: 2.8– 16.4 ng m$^{-3}$) was slightly lower than anthro-Fe concentration during the EAout period, excluding the lockdown period (avg ± 1σ: 8.2 ± 4.1 ng m$^{-3}$ ng m$^{-3}$, range: 0–14.0 ng m$^{-3}$). The reduction in anthro-Fe concentration by the limitation of human activities during the COVID-19 lockdown highlighted the importance of anthro-Fe as the source of Fe in fine aerosol particles. The $EF_{T\text{-}Fe}$ of the fine aerosol particles in the lockdown period (January 2019, $EF_{T\text{-}Fe}$: 0.45) were lower than those in the pre-

lockdown (December 2019, $EF_{T\text{-}Fe}$: 1.9) and post-lockdown (February 2020, $EF_{T\text{-}Fe}$: 1.4, Fig. 3b) periods. Similarly, the $EF_{T\text{-}Fe}$ in PM$_{2.5}$ during the lockdown in Hangzhou, China ($EF_{T\text{-}Fe}$: 1.6) was much lower than $EF_{T\text{-}Fe}$ in the pre-lockdown ($EF_{T\text{-}Fe}$: 13.3) and post-lockdown ($EF_{T\text{-}Fe}$: 6.6, Liu et al., 2021) periods. The decrease in $EF_{T\text{-}Fe}$ was attributed to the decrease in Fe-rich particles emitted from non-exhaust vehicle sources (Li et al., 2022), which were emitted from the abrasion processes of brake rings and tire ware. Furthermore, Fe concentrations in PM$_{2.5}$ collected in Tangshan and Wuhan decreased because of the

reduction in anthropogenic emissions, including those from the steel industry (Zheng et al., 2020; Xu et al., 2022). Thus, anthro-Fe was one of the dominant sources of Fe in fine aerosol particles in East Asia.



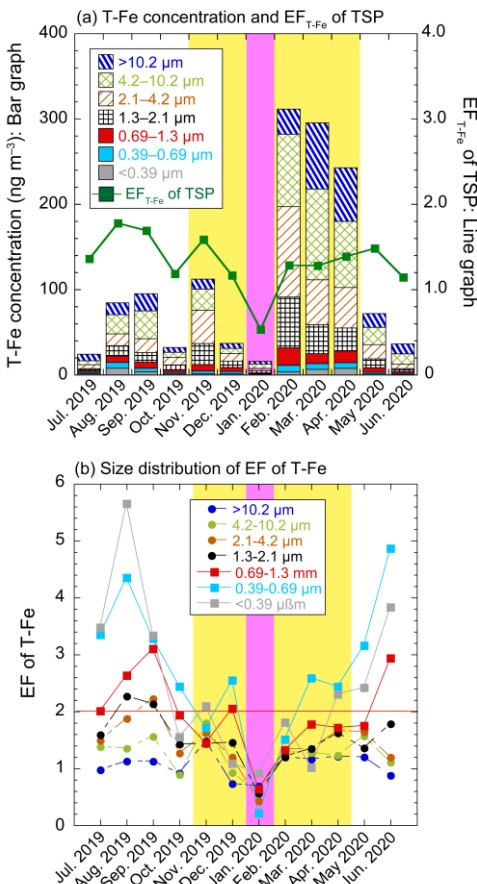

**Figure 3.** Monthly variations and size distributions of (a) T-Fe concentration in TSP, (b) $EF_{T-Fe}$ (red line: $EF_{T-Fe}$ is 2.0). The data of coarse aerosol particles are shown in dashed boxes or lines, while the data of fine aerosol particles are described in solid boxes or lines. Yellow and pink shaded regions show the EAout and COVID-19 lockdown periods, respectively.

### 3.3. Monthly variation and size distributions of d-Fe, $Fe_{sol}\%$, and [d-Fe]/[d-Al] ratio

#### 3.3.1.    Coarse aerosol particles

The d-Fe concentrations and $Fe_{sol}\%$ in TSPs varied from 0.6 to 14.7 ng m$^{-3}$ (avg ± 1σ: 6.3 ± 4.0 ng m$^{-3}$) and from 2.8% to 17.4% (avg ± 1σ: 8.3 ± 5.3%), respectively (Fig. 4a). The seasonal average $Fe_{sol}\%$ of TSP for the EAout periods (avg ± 1σ: 4.9 ± 4.3%) were lower than those for the JPN period (avg ± 1σ: 11.6 ± 4.2%). The $Fe_{sol}\%$ values in TSPs collected over the Pacific Ocean were typically 1.0–10%, consistent with those in our TSP samples collected during the EAout period (Table S3). The d-Fe concentration in the TSPs decreased from August 2019 to January 2020 and then increased from January to June 2020, consistent with previous observations in Japan (Fig. 4a; Sakata et al., 2023, Takahashi et al., 2013). The d-Fe concentration in TSP from July 2019 to January 2020 was controlled by factors that affect $Fe_{sol}\%$ (*e.g.*, emission sources and





chemical alteration of Fe-bearing particles) because monthly variations were similar between d-Fe concentration and $Fe_{sol}\%$ in the period (Fig. 4a). In contrast, the d-Fe concentrations in TSPs collected from February to April were considerably higher than the d-Fe concentration in the January sample, but the $Fe_{sol}\%$ values in TSPs collected from February to April were almost the same as that of the January sample (Fig. 4a). In this case, the atmospheric concentration of d-Fe increased because of the large loading of mineral dust in the atmosphere.


The $Fe_{sol}\%$ of coarse aerosol particles (avg $\pm$ 1σ: 2.2% $\pm$ 3.0%, range: 0.1%–13.6%) was slightly higher than that of typical mineral dust (<1%, Fig. 4b). The [d-Fe]/[d-Al] ratio in coarse aerosol particles (avg $\pm$ 1σ: 0.28 $\pm$ 0.12, range: 0.13–0.82) was consistent with d-Fe dissolved from Asian dust by proton-promoted dissolution (0.24 $\pm$ 0.20, Figs. 5a and 5b). This result indicates that d-Fe in coarse aerosol particles mainly originated from proton-promoted dissolution of mineral dust. Indeed, the correlation of $Fe_{sol}\%$ with [nss-$SO_4^{2-}$]/[T-Fe], further supports the contribution of proton-promoted dissolution to $Fe_{sol}\%$ in coarse aerosol particles (Fig. S5a). Furthermore, the $Fe_{sol}\%$ in the coarse aerosol particles increased with decreasing aerosol diameter because of increasing specific surface area, which is one of the factors controlling aerosol reactivity (Fig. 4b). A similar result was obtained by the observational study at Higashi-Hiroshima, Japan (Sakata et al., 2023).


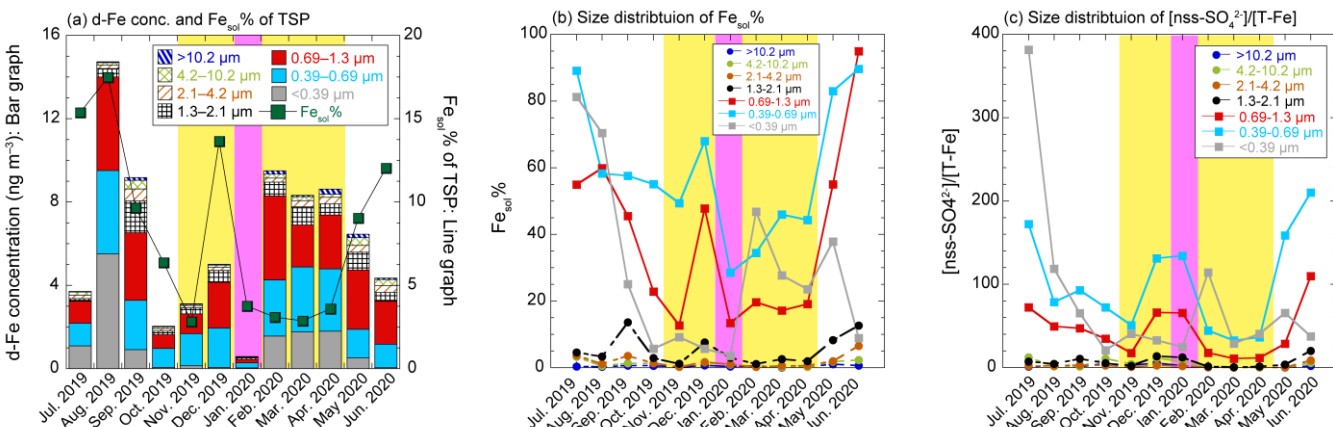

**Figure 4.** (a) d-Fe concentration and $Fe_{sol}\%$ in TSP (b) $Fe_{sol}\%$, and (c) [nss-$SO_4^{2-}$]/[T-Fe]. The data of coarse aerosol particles are shown in dashed boxes or lines, while the data of fine aerosol particles are described in solid boxes or lines. Yellow and pink shaded regions show the EAout and COVID-19 lockdown periods, respectively.

### 3.3.2. Fine aerosol particles

The summation of d-Fe in fine aerosol particles varied from 0.5 to 14.0 ng $m^{-3}$ (avg $\pm$ 1σ: 5.3 $\pm$ 3.7 ng $m^{-3}$), accounting for 71.1–94.8% (avg $\pm$ 1σ: 81.7 $\pm$ 7.0%) of d-Fe in the TSPs (Fig. 4a). The $Fe_{sol}\%$ in each size fraction of fine aerosol particles (avg $\pm$ 1σ: 42.1 $\pm$ 25.6%, range: 4.1–94.9%) was an order of magnitude higher than those in coarse aerosol particles (Fig. 4b). As mentioned above, the $Fe_{sol}\%$ in our TSP samples were identical to those in Pacific aerosol (Table S3), and the size distribution of $Fe_{sol}\%$ of our samples was consistent with previous observational studies conducted in East Asia and the Pacific

Ocean (Sakata et al., 2022, 2023; Kurisu et al., 2024). Thus, fine aerosol particles transported from East Asia play an essential





role in the supply of d-Fe to the North Pacific Ocean. Chemical alterations, including aerosol acidification, was one of the factors increasing $Fe_{sol}$% in fine aerosol particles because the $Fe_{sol}$% of fine aerosol correlated with the molar ratio of non-sea-salt sulfate to T-Fe ([nss-$SO_4^{2-}$]/[T-Fe]; Fig. S5b). Iron-bearing particles in fine aerosol particles were more acidified than coarse aerosol particles because the annual average of [nss-$SO_4^{2-}$]/[T-Fe] in fine aerosol particles (avg ± 1σ: 75 ± 71, range:

11–381) was higher than that in coarse aerosol particles (avg ± 1σ: 4 ± 5, Fig. 4c). The average [nss-$SO_4^{2-}$]/[T-Fe] ratio for the JPN period (avg ± 1σ: 101 ± 87) was higher than that for the EAout period (avg ± 1σ: 50 ± 39; Fig. 4c). This result indicated that fine aerosol particles collected for the JPN period were more acidified than those for the EAout period, consistent with higher $Fe_{sol}$% in the JPN period than in the EAout period.

The [d-Fe]/[d-Al] ratio in fine aerosol particles ranged from 0.14 to 3.78 (avg ± 1σ: 1.18 ± 0.77, Fig. 5a). Ligand-promoted

dissolution of mineral dust can increase [d-Fe]/[d-Al] ratio in aerosol particles, but this process was not the cause of the high [d-Fe]/[d-Al] ratio because of the absence of aerosol samples in the area (iv) in Fig. 5b. Although the [d-Fe]/[d-Al] ratio in fine aerosol particles was higher than coarse aerosol particles because of the influence of anthro-Fe with high [d-Fe]/[d-Al] ratio (Fig. 5b), the ratio reflected the values characteristic of mineral dust only during the COVID-19 period (pink diamonds in Fig. 5b). This result indicates that anthro-Fe is a dominant source of d-Fe under normal conditions. Furthermore, the data

for fine aerosol particles plotted along the mixing line between proton-promoted dissolution of mineral dust and highly soluble anthro-Fe indicate that these two processes are the dominant sources of d-Fe in fine aerosol particles (Fig. 5b). The contribution of d-Fe from highly soluble anthro-Fe was further supported by the correlation between $EF_{T-Fe}$ and $Fe_{sol}$% in fine aerosol particles (Fig. 5c).

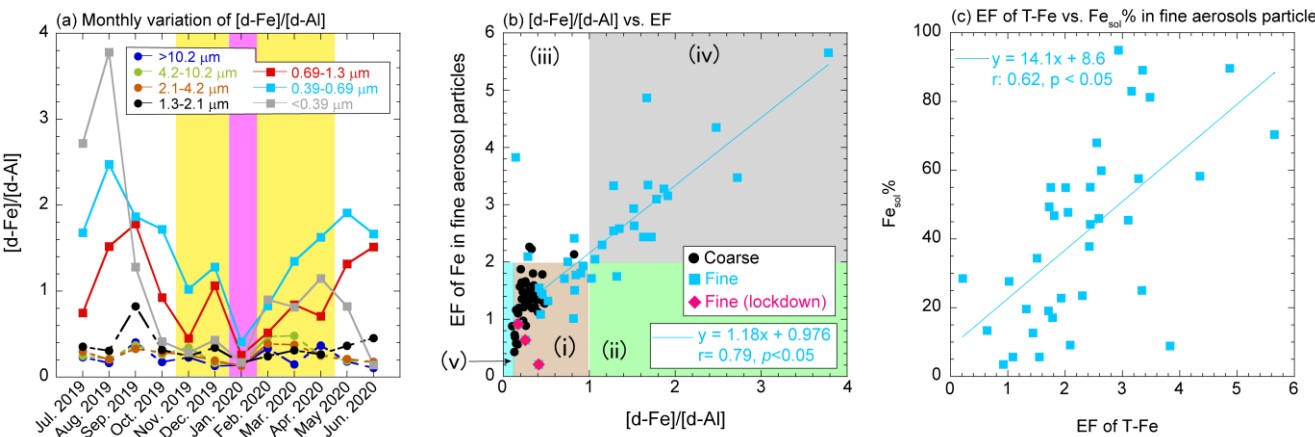

**Figure 5.** (a) A size distribution of [d-Fe]/[d-Al] ratio. The yellow and pink areas are shown in the JPN and COVID-19 lockdown periods. (b) relationships of $EF_{T-Fe}$ and [d-Fe]/[d-Al] ratio. The background color indicates the emission sources of T-Fe and d-Fe, which are detailed in Fig. 2. (c) a correlation between $EF_{T-Fe}$ and $Fe_{sol}$%.



The annual average of $F_{anthro}$ in fine aerosol particles was 46.2 ± 26.3% (range: 1.4–100%) and was higher during the JPN period than during the EAout period (Fig. 6a). The $F_{anthro}$ was most often the highest in the 0.39–0.69 µm fraction (Fig. 6a), consistent with the results from previous studies using the [d-Fe]/[d-Al] and Fe isotope ratios (Kurisu et al., 2016; Sakata et al., 2023). In TSPs, the seasonal average of $F_{anthro}$ values for the JPN and EAout periods were 33.7 ± 20.9% and 16.6 ± 9.6%, respectively. The lower $F_{anthro}$ in TSPs than fine aerosol particles attributed to large contribution of mineral dust in coarse aerosol particles, especially during EAout period. A similar result has been reported by a previous study performed in Higashi-Hiroshima, Japan, in 2013 (range: 1.48–80.7%, JPN: 29.4 ± 25.8%, EAout: 13.5 ± 10.6%, Sakata et al., 2023). Although annual anthro-Fe emissions in China are an order of magnitude higher than those in Japan (Kajino et al., 2020), the lower $F_{anthro}$ in the EAout period compared with the JPN period can be attributed to the large emission of mineral-Fe, especially in spring. Thus, mineral dust was the most dominant source of d-Fe in TSPs collected at the eastern end of East Asia, but the contribution of anthro-Fe to d-Fe cannot be negligible.

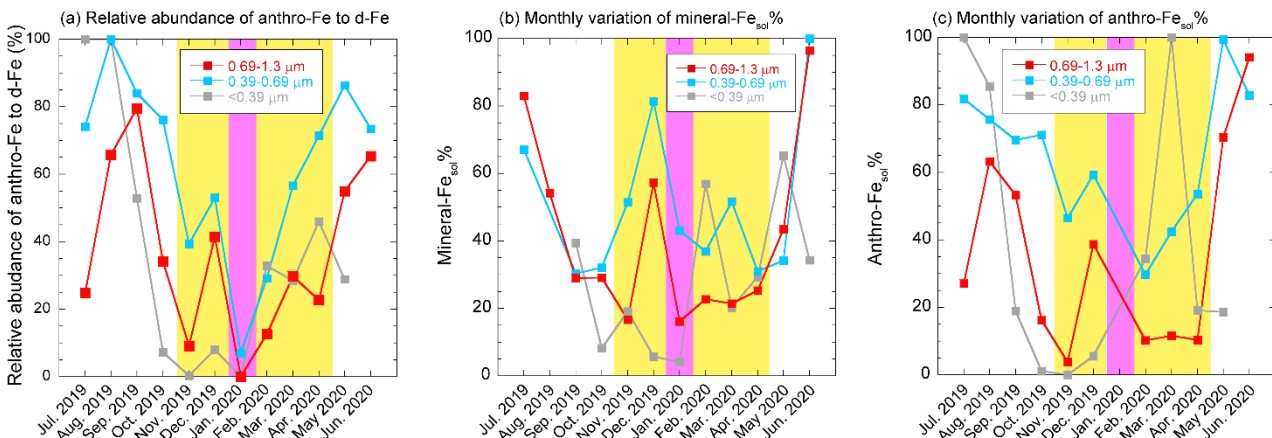

**Figure 6.** (a) A size distribution of [d-Fe]/[d-Al] ratio. (b) relationships of EF$_{T-Fe}$ and [d-Fe]/[d-Al] ratio. Background color indicates the major sources of T-Fe and d-Fe in aerosols. The (c) a correlation between EF$_{T-Fe}$ and Fe$_{sol}$%. (d–f) monthly trends of relative abundance of anthro-Fe to d-Fe, mineral-Fe$_{sol}$%, and anthro-Fe$_{sol}$% in fine aerosol particles, respectively. The yellow and pink areas are shown in the JPN and COVID-19 lockdown period.

### 3.4. Fe$_{sol}$% of mineral dust and anthropogenic aerosols

The annual average of mineral-Fe$_{sol}$% (40.5 ± 24.8%) was much higher than Fe$_{sol}$% of coarse aerosol particles (Fig. 6b). Considering that Fe$_{sol}$% of mineral dust at the emission is typically lower than 1% regardless of aerosol diameter (Shi et al., 2011b), the high mineral-Fe$_{sol}$% values observed in fine aerosol particles were caused by severe chemical alteration of mineral dust. Indeed, mineral-Fe$_{sol}$% correlated with [nss-SO$_4^{2-}$]/[T-Fe], which were plotted on an extension of the approximate line for coarse aerosol particles (Fig. S6a). This result indicated that mineral dust in fine aerosol particles underwent similar alteration processes to those in coarse aerosol particles, but the extent of chemical alteration differed between coarse and fine





aerosol particles. Despite mineral dust in fine aerosol particles exhibited high $Fe_{sol}$%, the annual average mineral-$Fe_{sol}$% in

TSP is only 4.4 ± 2.3% (range: 1.9–9.5%) owing to the low $Fe_{sol}$% of mineral dust in coarse aerosol particles. This finding

emphasizes the importance of the chemical alterations of mineral dust in fine aerosol particles for the d-Fe supply via mineral

dust deposition.

The annual average of anthro-$Fe_{sol}$% was 46.7 ± 32.9%, which was higher in the JPN period than in the EAout period

(Fig. 6c). The anthro-$Fe_{sol}$% can be enhanced not only by chemical alteration of anthropogenic aerosols but also by the direct

emission of highly soluble anthro-Fe emitted from liquid fuel combustions, including fuel oil and gasoline ($Fe_{sol}$%: up to 80%,

Sedwick et al., 2007; Sholkovitz et al., 2009; Schroth et al., 2009; Oakes et al., 2012). However, the contribution of anthro-Fe

from liquid fuel combustion to T-Fe and d-Fe in our samples was not significant, as described in the following section (Fig.

7). Therefore, the seasonal fluctuation in anthro-$Fe_{sol}$% is primarily controlled by the extent of chemical alterations of anthro-

Fe. This finding is supported by the strong correlation between anthro-$Fe_{sol}$% and [nss-$SO_4^{2-}$]/[T-Fe] (Fig. S6b). Notably,

anthro-$Fe_{sol}$% tended to be higher than mineral-$Fe_{sol}$% during the JPN period, whereas the opposite was true during the EAout

period, with mineral-$Fe_{sol}$% exceeding anthro-$Fe_{sol}$% (Figs. S6c–S6e). This shift is likely attributable to the differing

sensitivities of mineral-$Fe_{sol}$% and anthro-$Fe_{sol}$% to changes in aerosol acidity (further details are discussed in Section 3.8).

### 3.5. Sources apportionment of Fe in fine aerosol particles by PMF

### 3.5.1.    Sources of T-Fe and anthro-Fe

Six factors were identified as sources of fine aerosol particles during the JPN+EAout period: (1) sea spray aerosol and

less-aged mineral dust (hereafter and fresh dust), (2) aged mineral and road dust (aged dust), (3) the steel industry, (4) heavy

oil combustion, (5) the non-steel industry, and (6) secondary sulfate aerosol and dissolved metals formed through aerosol

acidification (secondary aerosol. Fig. S7). Here, dust includes mineral-Fe and anthro-Fe, such as non-exhaust vehicle particles

in road dust. Detailed classification methods, including tracer elements for each factor, are described in Supplemental

Information. It should be noted that several factors grouped multiple emission sources due to the similar emission processes

and/or physicochemical properties. For instance, sea spray aerosols and less-aged mineral dust in factor 1, both of which are

wind-blown from their sources, are likely to exhibit covarying atmospheric concentrations. Consequently, the PMF model may

have limitations in resolving covariant sources (Pindado and Perez, 2011).

Next, PMF analyses were performed individually using fine aerosol particles collected during the JPN and EAout periods

to evaluate the seasonal average contribution of each factor to T-Fe, anthro-Fe, and d-Fe (Fig. 7). Moreover, the $EF_{T-Fe}$, $Fe_{sol}$%,

and [d-Fe]/[d-Al] ratio of each factor were also estimated by PMF for each period (Tables S4 and S5). The same factors were

identified as the dominant sources of fine aerosol particles during the JPN period (Fig. S8). While heavy oil combustion was

not identified as a significant source of fine saerosol particles during the EAout period, the other five emission sources

remained important contributors to the source of fine aerosol particles in this period. (Fig. S9). These results are reasonable

because the PMF analysis of the JPN+EAout period showed a small contribution to the heavy oil combustion during the EAout

period (Fig. S7d). T-Fe in fine aerosol particles during the JPN and EAout periods were mainly derived from the steel industry




followed by aged dust, fresh dust, and secondary aerosol (Figs. 7a and 7d). Anthro-Fe in fine aerosol particles collected during the JPN period originated from the steel industry (36.8%) and secondary aerosols associated with high-temperature combustion

(27.1%, Fig. 7b). Given that sulfur dioxide, a precursor of sulfate aerosols in East Asia, was mainly emitted from coal combustions (Wang et al., 2014; Kurokawa and Ohara, 2020), the anthro-Fe in the secondary aerosol factor was also emitted from coal combustion. Non-exhaust vehicle particles in aged and fresh dusts contributed an anthro-Fe sources (28.5%, Fig. 7b). Thus, anthro-Fe in fine aerosol particles originated from high-temperature combustions and non-combusted anthro-Fe. This result is consistent with anthro-Fe source in Japanese PM$_{2.5}$ estimated by a semi-bottom-up model (Kajino et al., 2020):

the steel industry (20–50%), brake pad debris (20–40%, main components of non-exhaust vehicle particles), and coal-fired power plants (10–20%). During the EAout period, approximately 80% and 20% of anthro-Fe originated from the steel industry and non-exhaust vehicle particles in aged dust + fresh dust factors, respectively (Fig. 7e). This result is consistent with the results of previous studies because approximately 90% and 60% of anthropogenic nanoparticles (mainly composed of magnetite) and anthro-Fe were emitted from the steel industry in China, respectively (Li et al., 2021; Chen et al., 2021). The

importance of the steel industry as the source of anthro-Fe was emphasized by the reduction of human activities by the COVID-19 lockdown because the normalized contribution of the steel industry in the lockdown period was considerably low compared with that in pre- and post-lockdown periods (Fig. S7c).

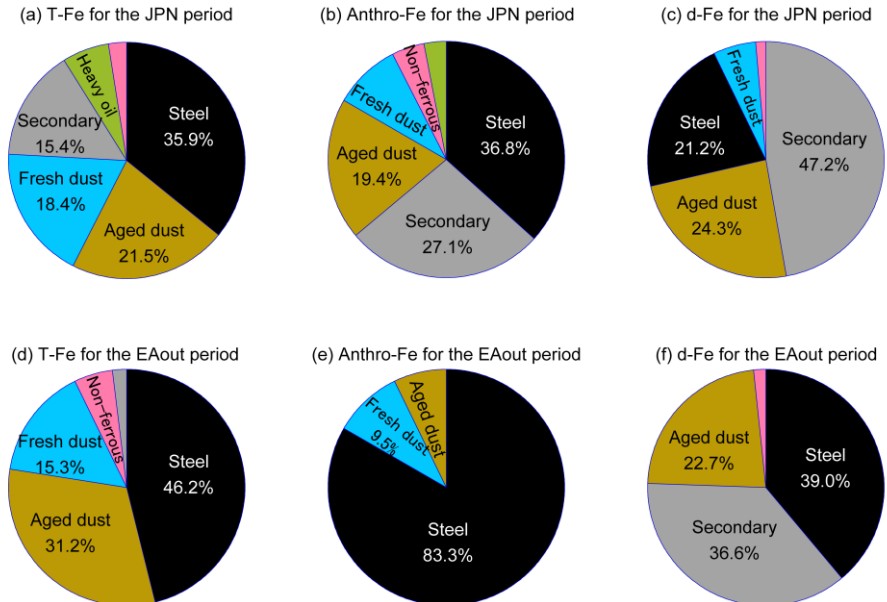

**Figure 7.** The average contribution of the emission sources to (a) T-Fe, (b)-anthro-Fe, and (c) d-Fe in fine aerosol particles
465       collected for the JPN period. (d-f) the same figures during the EAout period.

### 3.5.2. Sources of d-Fe



PMF analysis indicated that d-Fe in fine aerosol particles collected during the JPN and EAout periods originated from three primary sources: the steel industry, aged dust, and secondary aerosol formation (Figs. 7c and 7f). The considerable contribution

of the secondary aerosol factor to d-Fe highlights the importance of aerosol acidification in the dissolution of Fe from fine aerosol particles. As mentioned previously, the secondary aerosol factor is significantly influenced by coal combustion, a primary source of $SO_2$, and this can be a source of d-Fe in aerosols. However, d-Fe in the factors likely originated from not only coal combustions but also d-Fe dissolved from other factors (*e.g.*, aged dust and the steel industry) because the d-Fe concentrations within the secondary aerosol factor exceeded T-Fe concentrations (Tables S4 and S5). The reason is that PMF

methods is unable to distinguish between direct d-Fe emissions from coal combustion and d-Fe dissolution from aerosols acidified by $SO_2$ emitted from coal combustion due to the covariance of d-Fe concentration with the nss-$SO_4^{2-}$ concentration, a limitation also observed in previous studies (Zhu et al., 2022; Gao et al., 2024; Sun et al., 2024). The contribution of d-Fe in the secondary aerosol in this study was higher than those reported for fine aerosol particles collected in urban areas in China because of the further chemical alterations of Fe-bearing particles during transport from China to Japan (Zhu et al., 2022; Gao

et al., 2024; Sun et al., 2024). Furthermore, the $Fe_{sol}$% of aged dust (JPN: 60.6%, EAout: 19.3%) and steel industry (JPN: 31.8%, EAout: 22.4%) were higher than those at the emissions (Ito et al., 2021; Li et al., 2017) because a part of d-Fe dissolved by the chemical alteration of Fe was included in these factors. Thus, PMF analysis showed that atmospheric processes of mineral dust and anthro-Fe play an important role in the source of d-Fe in fine aerosol particles.

The PMF-estimated [d-Fe]/[d-Al] ratios for the steel industry were 5.67 and 1.20 for the JPN and EAout periods,

respectively. This result indicated that the high [d-Fe]/[d-Al] ratio in fine aerosol particles was mainly attributed to the d-Fe dissolved from anthro-Fe emitted from the steel industry. By contrast, the [d-Fe]/[d-Al] ratio in the factor of aged dust (JPN: 0.92, EAout: 0.69) was within the range of mineral dust of proton-promoted dissolution, but the ratio was higher than the average ratio for coarse aerosol particles less influenced by aerosol acidification (= 0.28 ± 0.12). As mentioned above, the aged dust fraction contained non-exhaust vehicle emissions (*e.g.*, brake rings and tire wear debris). Given that the $Fe_{sol}$% values

of brake ring and tire wear debris were less than 0.01% without chemical alterations (Shupert et al., 2013; Halle et al., 2021), the increase in [d-Fe]/[d-Al] ratio in the factor may have been caused by Fe dissolution from these materials during chemical alterations in the atmosphere. Thus, anthro-Fe emitted from high-temperature combustions and non-vehicle exhaust particles (*i.e.*, non-combusted anthro-Fe) contributed as a source of d-Fe in the fine aerosol particles.

### 3.6. Monthly variation and size distributions of Fe species

The abundances of Fe species in size-fractionated aerosol particles were estimated through the linear combination fitting of the XANES spectra of aerosol samples with those of reference materials (Fig. S10). Representative Fe species in coarse aerosol particles were ferrihydrite and Fe in crystalline aluminosilicates (*e.g.*, illite, biotite, and smectite; Figs. S10 and S11), which were similar to the species in mineral dust (Jeong and Achterberg, 2014; Jeong, 2020). Spot analyses of Fe species in

coarse aerosol particles revealed that Fe species in most measurement spots were consistent with Fe species in coarse aerosol particles detected through macroscopic XANES spectroscopy (Fig S12a). The sulfur intensity of these measurement spots was



not intense (white circle in Fig. S12d), indicating that less-aged mineral dust was dominant in the spots. By contrast, Fe(II)-sulfate and Fe(III)-sulfate coexisted with aluminosilicate and Fe-(hydr)oxides in spots with high sulfur intensity (green circle in Figs. S12b–S12d). This result indicated that Fe(II)- and Fe(III)-sulfates were present in severely aged mineral dust in the

coarse aerosol particles. Fe(II)- and Fe(III)-sulfates are water-soluble Fe species, which can enhance $Fe_{sol}$% in aerosol particles. However, the effects of Fe(II)- and Fe(III)-sulfate on $Fe_{sol}$% in coarse aerosol particles were not substantial because the abundance of Fe-sulfates to T-Fe was below the detection limit for macroscopic XANES (Fe-sulfates/T-Fe <10%), consistent with the low $Fe_{sol}$% in coarse aerosol particles.

Fe(II)-sulfate, Fe(III)-sulfate, and Fe(III)-oxalate were found as representative Fe species in fine aerosol particles (Fig. 8a).

Iron(III)-oxalate is also known as water-soluble Fe species. The most important result is that the abundance of these water-soluble Fe species is correlated with the $Fe_{sol}$% in fine aerosol particles (Fig. 8b). To confirm whether these water-soluble Fe species were readily dissolved in water, the Fe species in the residue of ultrapure water extraction (*i.e.*, insoluble Fe species) were determined. As a result, crystalline aluminosilicates and Fe oxides (hematite and magnetite) were found as insoluble Fe species in the residues, whereas Fe(II)-sulfate, Fe(III)-sulfate, and Fe(III)-oxalate were not detected (Fig. S13). Thus, the

$Fe_{sol}$% in fine aerosol particles were strongly related to the abundance of water-soluble Fe species. These water-soluble Fe species were derived from either or both direct emissions from high-temperature combustion and secondary formation in the atmosphere. Although Fe(II)- and Fe(III)-sulfates are directly emitted from liquid fuel combustion (Schroth et al., 2009; Oakes et al., 2012), these emissions were not identified by the PMF analysis as the dominant source of Fe in fine aerosol particles (Figs. 7c and 7f). Furthermore, Fe(III)-oxalate has not been detected from the emission source samples of anthro-Fe. Therefore,

these water-soluble Fe species were likely formed by the chemical alterations of the Fe in the fine aerosol particles. Aerosol samples in 0.39–0.69 and 0.69–1.3 µm fractions contained at least one of the water-soluble Fe species throughout the sampling campaign, whereas the finest fraction did not always contain these water-soluble Fe species (Fig. 5a). These results indicated that the degree and process of chemical alterations differs between the finest fraction and the 0.39–0.69 and 0.69–1.3 µm fractions. Previous studies showed that bared Fe-rich particles (= uncoated with sulfate and oxalate) were mainly present in

particles finer than 0.4 µm, which was expected to be less aged by atmospheric processes (Zhu et al., 2020; 2022, Xu et al., 2023). By contrast, Fe-rich particles coated with sulfate and oxalate were approximately 0.6 µm (Zhang et al., 2019; Zhou et al., 2020; Zhu et al., 2020, 2022; Xu et al., 2023). Sulfate and oxalate, mainly formed through chemical reactions in cloud water, are abundant in fine aerosol particles around $0.7 \pm 0.2$ µm (John et al., 1990; Meng and Seinfeld, 1994; Yu et al., 2005; Zhang et al., 2017). This diameter was consistent with those of Fe-bearing particles mixed internally with sulfate and oxalate, which were one of the components of cloud interstitial particles with a typical diameter of 0.5–1.0 µm (Zhang et al., 2017; Li

et al., 2013; Liu et al., 2018). Thus, the internal mixing of Fe-bearing particles with sulfate and oxalate was promoted in the cloud water and interstitial cloud particles.



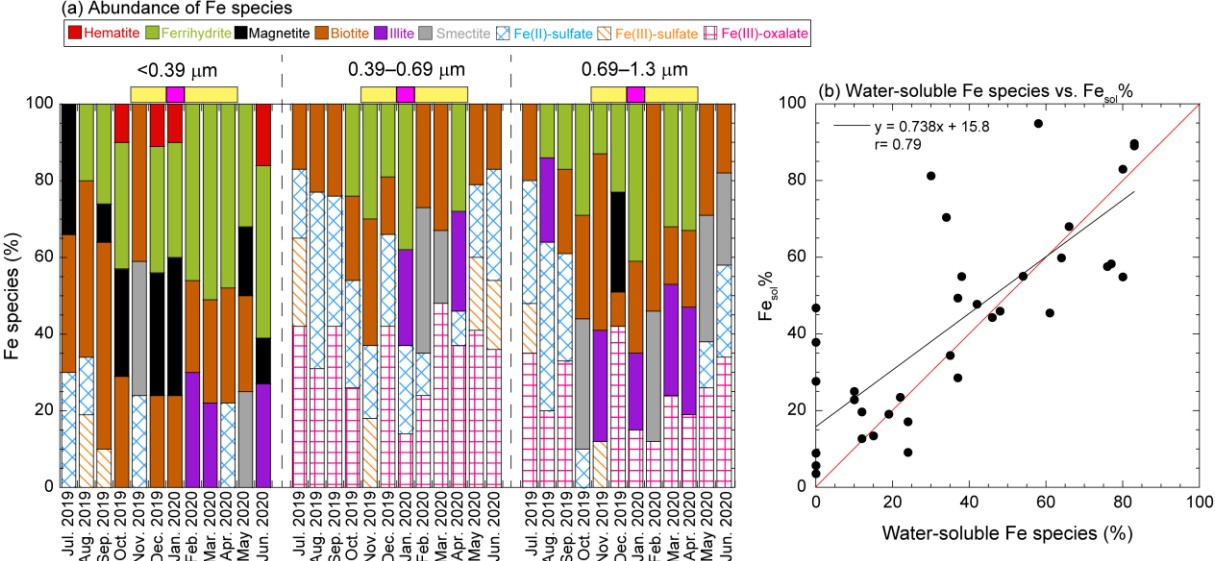

**Figure 8.** (a) Representative Fe species and their abundances in fine aerosol particles. Water-soluble Fe species (Fe(II)-sulfate, Fe(III)-sulfate, and Fe(III)-oxalate) were shown with lattice patterns. Yellow and pink bars above the panels show the period of Asian outflow and the COVID-19 lockdown, respectively. (b) A scatter plot between the abundance of water-soluble Fe species and $Fe_{sol}\%$ in fine aerosol particles.

### 3.7. Alteration processes and dissolution pH of mineral dust

To assess the alteration process of mineral-Fe and anthro-Fe in fine aerosol particles, we determined the Fe species of mineral dust and anthropogenic aerosol in September 2019 were through μ-XRF-XANES. The technique is suitable for source identification of metal elements in aerosol particles through the determination of elemental compositions and chemical species in regions of interest (Sakata et al., 2017, 2021). The regions of interest for this study were Fe-poor spots (M1–M10) and Fe-rich spots (A1–A12, Figs. 9a and S14). The Fe-poor spots contained Ca but did not contain anthropogenic metals (*e.g.,* Mn, Ni, Cu, Zn, and Pb, Figs. 9a and S14), indicating that Fe in these spots was associated with mineral dust. The M1 spot contained less aged mineral dust because (i) the XRF spectrum of the spot did not yield an intense S peak and (ii) Fe species in the spot (aluminosilicates and hematite) were similar to mineral dust (Figs. 9a and 9b). By contrast, Fe(II)- and Fe(III)-sulfates coexisted with aluminosilicates in M2–M10 spots, and the XRF spectra yielded an intense peak of sulfur. These results showed that internal mixing of Fe-bearing aluminosilicates with sulfate is important to the formation of Fe(II)- and Fe(III)-sulfates in the atmosphere. The average abundance of water-soluble Fe species in the Fe-poor spot (avg ± 1σ: 46 ± 25%) was higher than that in mineral-$Fe_{sol}\%$ estimated by Eq. 3 (20.3%). This result is partly due to the small number of measurements of Fe species at points of low S intensity, such as the M1 spot.

Given that Fe(III)-oxalate was detected in mineral dust in fine aerosol particles (Fig. 9a), ligand-promoted dissolution appears to contribute to Fe dissolution from mineral dust. Previous research has shown that oxalate plays two key roles in





controlling $Fe_{sol}\%$ in aerosols, depending on the aerosol acidity (Myriokefalitakis et al., 2015; Tao and Murphy, 2019; Sakata et al., 2022; Zhang et al., 2024). The first role of oxalate in Fe dissolution from mineral dust is to stabilize d-Fe in the aqueous phase as oxalate complexes after proton-promoted dissolution under highly acidic conditions (pH < 3.0). In such a pH conditions, oxalate does not significantly contribute to Fe release from mineral dust because the dissolution rate of Fe from
aluminosilicate mineral via proton-promoted dissolution is more than an order of magnitude higher than that via ligand-promoted dissolution (Balland et al., 2010; Cappelli et al., 2020). Additionally, Fe(III)-oxalate can be stabilized under highly acidic solutions (Sakata et al., 2022), which not only stabilizes Fe in the aqueous phase but may also promote further Fe dissolution by proton-promoted dissolution by reducing the saturation of inorganic Fe (Ito and Shi, 2016). Thus, oxalate assists the Fe dissolution from mineral dust via proton-promoted dissolution under highly acidic conditions. Conversely,  the
dissolution rate of Fe from aluminosilicate via ligand-promoted dissolution exceeds those via proton-promoted dissolutions under moderately acidic conditions (pH > 3.0) because the dissolution rate of Fe from mineral dust by proton-promoted dissolution decreases markedly with increasing pH. Therefore, second role of oxalate is the promotion of Fe dissolution from aluminosilicate under moderately acidic conditions, but its effect is not sufficient to dissolve as much Fe from mineral dust as proton-promoted dissolution under highly acidic conditions (Balland et al., 2010). Indeed, previous experiments on the ligand-
promoted dissolution of mineral dust in simulated cloud water showed that organic ligands, such as oxalates (0–8 μmol L$^{-1}$), increased $Fe_{sol}\%$ but was limited to less than 1% (Paris et al., 2011, Paris and Desboeufs, 2013). Therefore, it is considered that Fe dissolution under highly acidic conditions is necessary to reach high mineral-$Fe_{sol}\%$ in the fine aerosol particles.

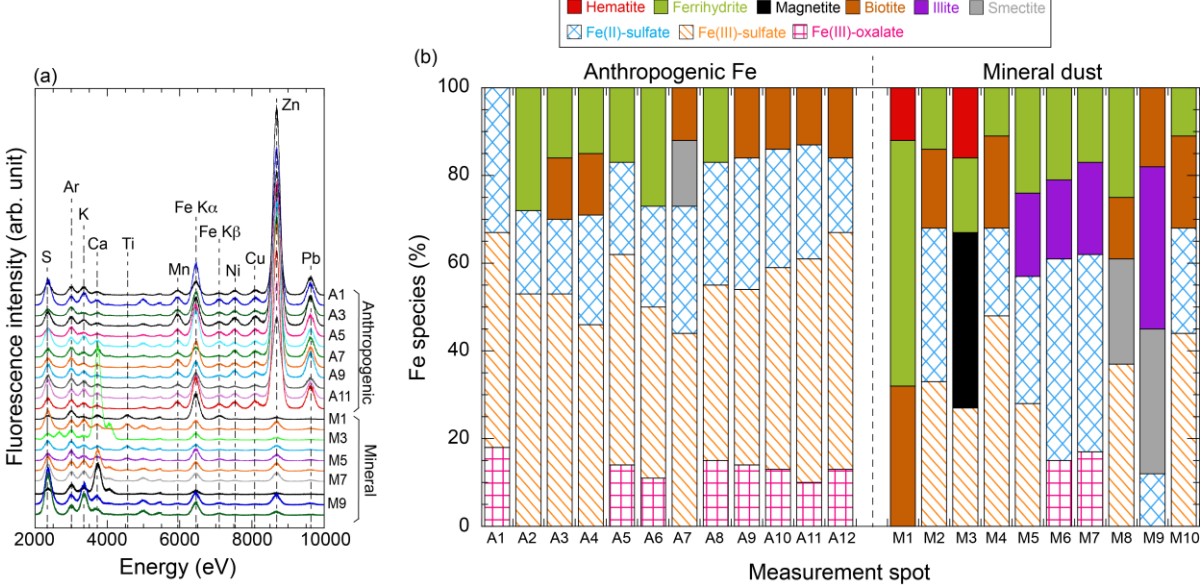

**Figure 9.** (a) μ-XRF spectrum of Fe-rich (anthropogenic: A1–A12) and Fe-poor (mineral: M1–M9) spots in 0.39–0.69 μm
575        aerosol particles collected in September 2019. (b) Abundance of Fe species in each measurement spot.



To assess whether mineral dust has undergone highly acidic conditions in the atmosphere, we estimated the aerosol pH of mineral dust ($pH_{mineral}$) to reach the observed mineral-$Fe_{sol}$% in the fine aerosol particles, assuming only proton-promoted dissolution. As a result, the average $pH_{mineral}$ during the JPN-period (0.60) was lower than that during the EAout-period
(average $pH_{mineral}$: 1.78; Fig. 10a). The decrease in aerosol pH during summer, as also indicated by aerosol pH estimation using thermodynamic models, can be attributed largely to the enhanced proton activity resulting from higher temperatures (Pye et al., 2020; Song and Osada, 2020). Thus, the seasonal variation in $pH_{mineral}$ is likely to be synchronized with the overall changes in aerosol pH. One potential issue is the mitigation of the decrease in $pH_{mineral}$ of mineral dust in fine aerosol particles due to the buffering capacity of alkaline minerals, including calcium carbonate ($CaCO_3$). Previous studies have shown that the
buffering capacity of alkaline mineral species in fine aerosol particles is almost completely consumed by chemical reactions with H2SO4, leading to the formation of $CaSO_4 \cdot 2H_2O$ during transport from East Asia to Japan (Takahashi et al., 2008; Miyamoto et al., 2020). Additionally, the thermodynamic model predicted that the pH of mineral dust would reach 0.0–1.0 after overwhelming the buffer capacity (Meskhidze et al., 2003, 2005). Therefore, it is reasonable to infer that the mineral dust in the fine aerosol particles underwent significant acidification (pH < 2.0). From these results, Fe(III)-oxalate was formed as a
result of the stabilization of d-Fe in the aqueous phase follwoing proton-promoted dissolution. This finding is supported by the lack of correlation of mineral-$Fe_{sol}$% with the abundance of Fe(III)-oxalate (Fig. 11a). The impoortance of the proton-promoted dissolution for the Fe dissolution from mineral dust in fine aerosol particles is consistent with our previous studies because Fe in mineral dust collected above the North Pacific Ocean ([d-Fe]/[d-Al]: 0.255–0.567) was dissolved by proton-promoted dissolution under highly acidic conditions, even though organic Fe complexes with humic-like substances (Fe(III)-
HULIS) are dominant Fe species in fine aerosol particles (Sakata et al., 2022, 2023).

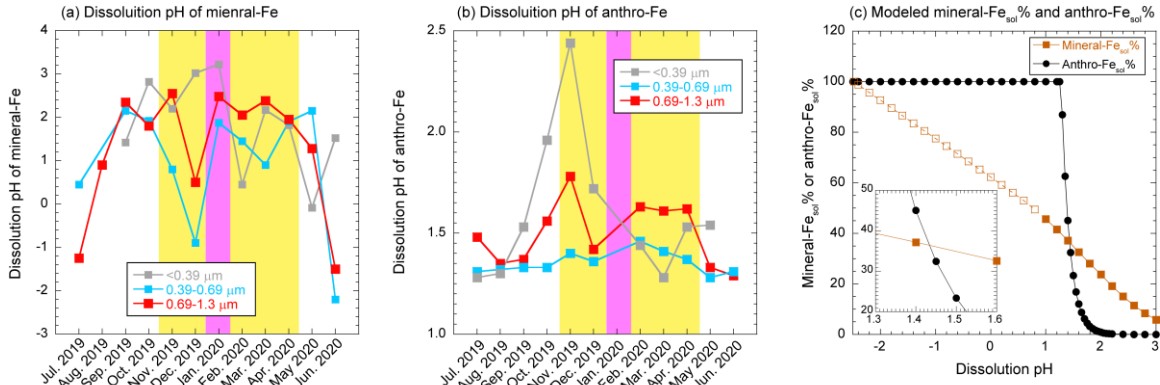

**Figure 10.** Monthly variation of (a) mineral-$Fe_{sol}$% and (b) anthro-$Fe_{sol}$% in fine aerosol particles. Yellow and pink shaded areas show the EAout and COVID-19 lockdown period, respectively. (c) pH dependences of modeled mineral-$Fe_{sol}$% and anthro-$Fe_{sol}$%. Mineral-$Fe_{sol}$% plotted with closed symbols was estimated using the kinetic data shown in Fig.




S15a. Mineral-Fe$_{sol}$% with open symbol was calculated by extrapolating the kinetic equation for pH 1–2 shown in Fig. S15a.

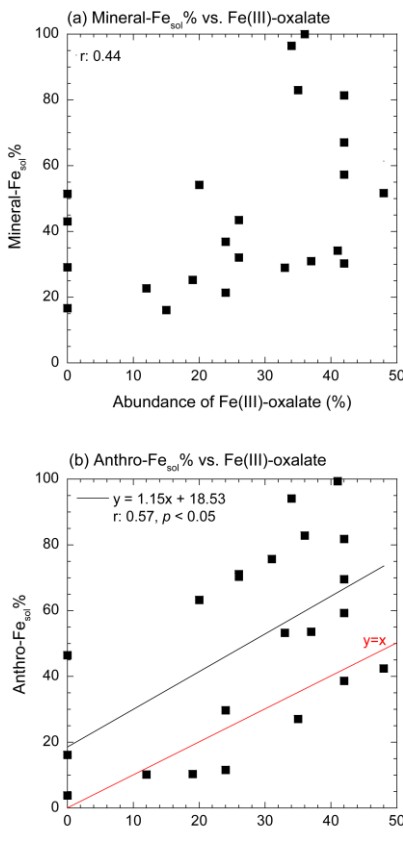

**Figure 11.** Scatter plots of abundance of Fe(III)-oxalate with (a) mineral-Fe$_{sol}$% and (b) anthro-Fe$_{sol}$% in fine aerosol particles (0.39–0.69 and 0.69–1.3 μm). The back and red lines in a panel (b) show regression line and y = x, respectively.

### 3.8. Alteration processes of anthro-Fe

Aerosol particles in Fe-rich spots primarily originated from anthropogenic emissions with high-temperature combustion because anthropogenic elements, including Mn, Ni, Cu, Zn, and Pb, were abundant in the spots (A1–A12 in Figs. 9a and S14). Furthermore, Fe intensities was much higher than those in Fe-poor spots containing mineral dust. This result suggests that the Fe in these spots originated from anthropogenic emissions with high EF$_{T-Fe}$, including the steel industry (Table S3 and S4). The high sulfur intensity of Fe-rich spots indicated that the anthro-Fe in these spots was significantly aged in the atmosphere (Figs. 6a and S14a). As evidence, more than half of the Fe in these spots existed as water-soluble Fe species, including Fe(II)-sulfate, Fe(III)-sulfate, and Fe(III)-oxalate (Fig. 9b). The average abundance of water-soluble Fe species in Fe-rich spots (avg ± 1σ: 81 ± 9%) was consistent with the anthro-Fe$_{sol}$% in the sample estimated by Eq. 4 (74.0%), indicating that the




representative anthro-Fe species in this sample can be determined. Furthermore, the consistency supports the reliability of estimating anthro-Fe$_{sol}$% based on the [d-Fe]/[d-Al] ratio can provide reasonable results.

The number of particles containing Fe(III)-oxalate appeared to be greater in Fe-rich spots than in Fe-poor spots (Fig. 9b).

The abundance of Fe(III)-oxalate in fine aerosol particles of 0.39–0.69 and 0.69–1.3 μm weakly correlated with anthro-Fe$_{sol}$% (r: 0.57; Fig. 11b). Therefore, oxalate may partially contribute to the dissolution of anthro-Fe. Oxalate in fine aerosol particles was formed in cloud water (average pH of East Asia: 4.2, Shah et al., 2020), increasing the number of oxalate-bearing Fe-rich particles through cloud processing (Li et al., 2013; Zhang et al., 2017; Liu et al., 2018). The acquisition of an oxalate coating by anthro-Fe in cloud water promoted Fe dissolution after anthro-Fe was released by cloud water evaporation because the

dissolution rate of oxalate-coated hematite at pH 2.4 is higher than that of noncoated hematite (Xu and Gao, 2008). By contrast, proton-promoted dissolution dominated Fe dissolution from hematite under highly acidic conditions (pH < 2.0, Xu and Gao, 2008). Assuming that anthro-Fe dissolution occurred solely through proton-promoted dissolution, the pH range for proton-promoted dissolution of anthro-Fe was estimated based on the solubility product of hematite nanoparticles. As a result, the predicted pH range for proton-promoted dissolution of anthro-Fe was narrow (1.3–2.0) due to the sharp increase in anthro-

Fe$_{sol}$% below pH 2.0 (Figs. 11b and 11c). Considering that anthro-Fe has underwent highly acidic conditions and the abundance of Fe(III)-oxalate is lower than anthro-Fesol%, it is inferred that proton-promoted dissolution was the primary mechanism for the dissolution of anthro-Fe. Subsequent complexation with oxalate in the aqueous phase to form Fe(III)-oxalate likely reduced the saturation index of inorganic Fe, potentially facilitating further proton-promoted dissolution from the solid phase (Ito and Shi, 2016).

As previously mentioned, anthro-Fe$_{sol}$% tend to be higher than mineral-Fe$_{sol}$% for the JPN period, while the opposite trend was observed for the EAout period (Figs. S6c–S6e). The seasonal trends of mineral-Fe$_{sol}$% and anthro-Fe$_{sol}$% can be explained by their responsiveness to dissolution pH, as both mineral-Fe and anthro-Fe were mainly dissolved by proton-promoted dissolution. Anthro-Fe$_{sol}$% exhibited a rapid increase from near 0% to 100% between pH 2.2 and 1.2, whereas mineral-Fe$_{sol}$% showed a gradual increase with decreasing pH. As a result, anthro-Fe$_{sol}$% exceeded mineral-Fe$_{sol}$% within the

pH range of 1.4 to 1.5 (Fig. 11c). Indeed, the pH of samples where anthro-Fe$_{sol}$% exceeded mineral-Fe$_{sol}$% during the JPN period was significantly lower than 1.4 (average pH: 0.60), while samples with higher mineral-Fe$_{sol}$% during the EAout period had a pH exceeding 1.4 (average pH: 1.78).

## 4. Implications

The dissolution of mineral-Fe and anthro-Fe in fine aerosol particles by atmospheric processes (*e.g.*, proton- and ligand-promoted dissolutions) during aerosol transport from East Asia to Japan plays a crucial role in supplying of d-Fe to the North Pacific. The Fe$_{sol}$% in TSP (2.8–17.4%) and PM$_{2.5}$ (8.0–29.2%) and size distributions in our samples were consistent with those of aerosol particles above the North Pacific (Table S2). In addition, the $F_{anthro}$ of TSP collected for the EAout period (20.4–43.9%) was consistent with the contribution of anthro-Fe to d-Fe in the North Pacific surface seawater considering the

influence of Asian aerosol deposition (21–59%, Pinedo-González et al., 2020). The consistencies of Fe$_{sol}$% and $F_{anthro}$ between





Japanese aerosols and North Pacific aerosols indicate that these parameters did not change drastically during transport from Japan to the North Pacific. This assumption is consistent with the results of previous studies (Buck et al., 2013; Sakata et al., 2022). Therefore, the long-term observation of factors controlling mineral-$Fe_{sol}$% and anthro-$Fe_{sol}$% in aerosol particles in the eastern end of East Asia leads to the development of our knowledge about Fe supply to the North Pacific Ocean via aerosol

deposition.

Although this study focused primarily on $Fe_{sol}$%, changes in Fe species during transport can affect the optical properties of Fe-bearing particles. Recently, along with black carbon (BC), anthro-Fe has been recognized as light-absorbing aerosol particles (Moteki et al., 2017; Ito et al., 2018; Lamb et al., 2021). The contributions of anthro-Fe on direct radiative forcing (DRF) relative to that for BC are approximately 10% in the polluted region and up to 6% in a remote area, including the marine

atmosphere (Moteki et al., 2017; Ito et al., 2018; Lamb et al., 2021). Surface coating (*e.g.*, sulfate) on BC and anthro-Fe can enhance light absorption by the lensing effect (Bond et al., 2006; Moteki et al., 2017; Liu et al., 2017). Unlike BC, internal mixing of anthro-Fe with sulfate or oxalate may produce less light absorbable Fe(II)-sulfate, Fe(III)-sulfate, or Fe(III)-oxalate, reducing the DRF of anthro-Fe. Indeed, a previous study has predicted that the transformation of anthro-Fe to ferrihydrite will reduce its DRF (Ito et al., 2018). Therefore, incorporating the transformation process of anthro-Fe to evaluate DRF (especially

in remote areas) is necessary because of anthro-Fe in climate models. Thus, atmospheric processes on Fe-bearing particles in East Asia affect climate regulation factors associated with Fe aerosol, especially ocean Fe fertilization and DRF of Fe oxides. Therefore, Fe speciation not only in emission source regions and the marine atmosphere but also at intermediate points along the transport pathway plays essential roles in constraining the control factors of $Fe_{sol}$% and DRF of Fe-bearing particles.




*Data Availability.*

All quantitative data is approval in ERAN database at https://www.ied.tsukuba.ac.jp/database (doi: 10.34355/CRiES.U.TSUKUBA.00157). The XAFS data are available upon request.

*Author Contributions.* K.S, S.T., A.M., H.T., and Y.T. designed the research. K.S., A.S., and A.M. collected size-fractionated aerosol samples. S.T determined trace metal concentrations. K.S, Y.T, and M.K. performed macroscopic and micro-focused XAFS experiments. K.S and Y.T. wrote the manuscript, and all authors approved the manuscript before submission.

*Competing interests.*

The authors declare that they have no conflict of interest.

**Acknowledgments**

We thank Haruka Naya and Megumi Matsumoto for collecting the aerosol samples. Macroscopic and semi-microscopic XAFS experiments were performed under the approval of the Photon Factory Program Advisory Committee (Proposal No. 685 2019G093).

*Financial support.*

This study was supported by Cooperative Research Program of the Institute of Nature and Environmental Technology, Kanazawa University (Proposal No. 19002) and JSPS KAKENHI (Grant Numbers: 20K19957).






**Figure captions**

**Figure 1.** The sampling site (NOTOGRO) of size-fractionated aerosol sampling.

**Figure 2.** The diagram of EFT-Fe and [d-Fe]/[d-Al] ratio for evaluating T-Fe and d-Fe sources of aerosol particles.

**Figure 3.** Monthly variations and size distributions of (a) T-Fe concentration in TSP, (b) $EF_{T-Fe}$ (red line: $EF_{T-Fe}$ is 2.0). The data of coarse aerosol particles are shown in dashed boxes or lines, while the data of fine aerosol particles are described in solid boxes or lines. Yellow and pink shaded regions show the EAout and COVID-19 lockdown periods, respectively.

**Figure 4.** (a) d-Fe concentration and $Fe_{sol}$% in TSP (b) $Fe_{sol}$%, and (c) [nss-$SO_4^{2-}$]/[T-Fe]. The data of coarse aerosol particles are shown in dashed boxes or lines, while the data of fine aerosol particles are described in solid boxes or lines. Yellow and pink shaded regions show the EAout and COVID-19 lockdown periods, respectively.

**Figure 5.** (a) A size distribution of [d-Fe]/[d-Al] ratio. The yellow and pink areas are shown in the JPN and COVID-19 lockdown periods. (b) relationships of $EF_{T-Fe}$ and [d-Fe]/[d-Al] ratio. The background color indicates the emission sources of T-Fe and d-Fe, which are detailed in Fig. 2. (c) a correlation between $EF_{T-Fe}$ and $Fe_{sol}$%.

**Figure 6.** (a) A size distribution of [d-Fe]/[d-Al] ratio. (b) relationships of $EF_{T-Fe}$ and [d-Fe]/[d-Al] ratio. Background color indicates the major sources of T-Fe and d-Fe in aerosols. The (c) a correlation between $EF_{T-Fe}$ and $Fe_{sol}$%. (d–f) monthly trends of relative abundance of anthro-Fe to d-Fe, mineral-$Fe_{sol}$%, and anthro-$Fe_{sol}$% in fine aerosol particles, respectively. The yellow and pink areas are shown in the JPN and COVID-19 lockdown periods.

**Figure 7.** The average contribution of the emission sources to (a) T-Fe, (b)-anthro-Fe, and (c) d-Fe in fine aerosol particles collected for the JPN period. (d-f) the same figures for the EAout period.

**Figure 8.** (a) Representative Fe species and their abundances in fine aerosol particles. Water-soluble Fe species (Fe(II)-sulfate, Fe(III)-sulfate, and Fe(III)-oxalate) were shown with lattice patterns. Yellow and pink bars above the panels show the period of Asian outflow and the COVID-19 lockdown, respectively. (b) A scatter plot between the abundance of water-soluble Fe species and $Fe_{sol}$% in fine aerosol particles.

**Figure 9.** (a) μ-XRF spectrum of Fe-rich (anthropogenic: A1–A12) and Fe-poor (mineral: M1–M9) spots in 0.39–0.69 μm aerosol particles collected in September 2019. (b) Abundance of Fe species in each measurement spot.

**Figure 10.** Monthly variation of (a) mineral-$Fe_{sol}$% and (b) anthro-$Fe_{sol}$% in fine aerosol particles. Yellow and pink shaded areas show the EAout and COVID-19 lockdown period, respectively. (c) pH dependences of modeled mineral-$Fe_{sol}$% and anthro-$Fe_{sol}$%. Mineral-$Fe_{sol}$% plotted with closed symbols was estimated using the kinetic data shown in Fig. S15a. Mineral-$Fe_{sol}$% with open symbol was calculated by extrapolating the kinetic equation for pH 1–2 shown in Fig. S15a.

**Figure 11.** Scatter plots of abundance of Fe(III)-oxalate with (a) mineral-$Fe_{sol}$% and (b) anthro-$Fe_{sol}$% in fine aerosol particles (0.39–0.69 and 0.69–1.3 μm). The back and red lines in a panel (b) show regression line and y = x, respectively.





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
