# Peer review of "Atmospheric chemistry in East Asia determines the iron solubility of aerosol particles supplied to the North Pacific Ocean"

_EGUsphere, 2025_

## Author Comment (AC2)

**Reviewer 2**

| Reviewer's comments | Author's reply |
|---|---|
| This study presents measurements of size-resolved soluble iron (Fe) content in aerosol particles over the North Pacific Ocean, based on samples collected over a one-year period. The authors distinguish the samples by air mass origin at the sampling site, allowing for a discussion of the potential mechanisms and sources influencing iron solubility across different size fractions. While the study is generally well-structured and clearly written, several points require clarification or further elaboration before publication. These are outlined below. | We sincerely thank the reviewer for the time and effort you have put into this review. We have carefully revised the manuscript with full consideration of the reviewer's comments and suggestions. Responses to the reviewer's comments are in black and not indented; corresponding revisions in the manuscript are in red and indented. |
| Line 130: Please clarify why the extracted solution was evaporated to dryness. What proportion of organically bound soluble Fe is likely to be lost during this procedure? Additionally, why was the residue re-dissolved in 2% $HNO_3$? The use of nitric acid could potentially increase the dissolved iron content, thereby altering the measured solubility. This methodology seems unconventional and requires justification. | I apologies for the lack of clarity regarding the water extraction method of Fe in aerosol particles. After the water extraction, the aerosol filter is removed from the vial, and the solution is filtered through a PES filter. Therefore, no additional d-Fe leaches out from the aerosol during the evaporation to dryness or the subsequent dissolution of the residue in 2% $HNO_3$.

Dissolved Fe in aerosol particles was extracted with 2–4 mL of ultrapure water in a polypropylene centrifuge tube and horizontal shaking for one day. After being subjected to water extraction, the PTFE filter was removed from the vial, and the solution was filtered through a PES syringe filter. The filtrated solutions were evaporated to dryness. The evaporated residue was then redissolved in 2% $HNO_3$. |
| Line 159: What is the EFT-Fe value for the second group? Given that the third group is described as being the opposite of the first, it is unclear how the first and second groups can share the same EFT-Fe range. What differentiates them if not the EFT-Fe? | We found no difference in $EF_{T-Fe}$ given that T-Fe in groups (i) and (ii) primarily originated from mineral particles. However, these groups were distinguished by the different dissolution processes of d-Fe within mineral particles. While Fe in mineral dust in the first group was mainly dissolved through proton-promoted dissolution, that in the second group dissolved through ligand-promoted dissolution. In general, organic ligands exhibited higher stability constants with Fe than with Al. As a result, the [d-Fe]/[d-Al] ratio for ligand-promoted dissolution ([d-Fe]/[d-Al]>1.0) was higher than that for proton-promoted dissolution ([d-Fe]/[d-Al]: 0.1–1.0).
By contrast, aerosol particles in the first and third groups were characterized by differences |

in anthro-Fe. In the group (iii), T-Fe was affected by anthro-Fe with high T-Fe and T-Al. Consequently, the group (iii) exhibited high $EF_{T\text{-}Fe}$.

A diagram of [d-Fe]/[d-Al] ratios combined with $EF_{T\text{-}Fe}$ are useful tools for evaluating the sources and dissolution processes of d-Fe in aerosol particles because the $Fe_{sol}\%$ values of aerosol particles vary depending on the dominant sources of T-Fe and d-Fe (Sakata et al., 2023). T-Fe and d-Fe sources can be categorized into the following five groups (Fig. 2): T-Fe in groups (i) and (ii) originate from mineral dust with $EF_{T\text{-}Fe}$ < 2.0. The [d-Fe]/[d-Al] ratio varies depending on the different dissolution processes of mineral dust (i.e., proton- and ligand-promoted dissolution processes). The [d-Fe]/[d-Al] ratio for ligand-promoted dissolution ([d-Fe]/[d-Al] > 1.0) is higher than that for proton-promoted dissolution ([d-Fe]/[d-Al]: 0.1–1.0) because Fe is preferentially dissolved by organic ligands over Al. T-Fe in groups (iii) and (iv) is derived from anthro-Fe with $EF_{T\text{-}Fe}$ > 2.0. In group (iii), T-Fe is mainly derived from anthro-Fe, whereas d-Fe is derived from mineral dust because anthro-Fe is present in the form of insoluble Fe, which cannot affect the [d-Fe]/[d-Al] ratio of aerosol particles. By contrast, the anthro-Fe in group (iv) is highly soluble, and its high T-Fe/T-Al ratio (i.e., high $EF_{T\text{-}Fe}$) is retained upon dissolution, as reflected by its [d-Fe]/[d-Al] ratio. Consequently, aerosols in group (iv) exhibit high $EF_{T\text{-}Fe}$ and [d-Fe]/[d-Al]. However, distinguishing between proton- and ligand-promoted dissolutions is difficult because highly soluble anthro-Fe exhibits high [d-Fe]/[d-Al] ratios in both processes. Herein, anthro-Fe refers to anthropogenic Fe-rich particles that can increase the $EF_{T\text{-}Fe}$ emitted from not only high-temperature combustion (e.g., steel industry and coal combustion; Kajino et al., 2020; Ito et al., 2021), but also non-combusted anthro-Fe (e.g., non-exhaust vehicle particles, such as brake ring and tire wear debris; Sanderson et al., 2016; Li et al., 2022; Fu et al., 2023). Finally, group (v) consists of aluminosilicate glasses emitted from high-temperature combustion, including coal combustion and municipal solid waste incineration. These particles are characterized by low $EF_{T\text{-}Fe}$ values (<2.0) and [d-

| | |
|---|---|
| | Fe]/[d-Al] ratios (<0.1). A detailed description of these five classifications is presented in S.1.1 in Supplemental Information. |
| Lines 376–380: Please elaborate on the mechanisms that explain the higher iron solubility observed in fine particles when the d-Fe/d-Al ratio is high. A clearer explanation of the processes involved would strengthen the interpretation. | When the method for a diagram between $EF_{T\text{-}Fe}$ and [d-Fe]/[d-Al], the following sentence has been added. |
| | Factors potentially contributing to an increased [d-Fe]/[d-Al] ratio in aerosols include ligand-promoted Fe dissolution from mineral dust and the contribution of anthro-Fe to d-Fe. However, considering the absence of aerosol samples in area (iv) as illustrated in Fig. 5b, ligand-promoted dissolution was not the primary cause of the high [d-Fe]/[d-Al] ratio in fine aerosol particles. Therefore, the elevated [d-Fe]/[d-Al] ratio in fine aerosol particles is primarily attributed to the influence of anthro-Fe. Indeed, the data for fine aerosol particles plotted along the mixing line between proton-promoted dissolution of mineral dust and highly soluble anthro-Fe (Fig. 5b) indicate that these two processes are dominant sources of d-Fe. The significant contribution of d-Fe from highly soluble anthro-Fe was further supported by the correlation between $EF_{T\text{-}Fe}$ and $Fe_{sol}\%$ of fine aerosol particles (Fig. 5c). Furthermore, observations during the COVID-19 lockdown period provide crucial insights into the importance of anthro-Fe as a source of d-Fe under normal conditions. This is because the [d-Fe]/[d-Al] ratio in fine aerosol particles collected during the lockdown period was similar to that of mineral dust (pink diamonds in Fig. 5b), suggesting a reduced influence of anthropogenic sources on d-Fe during the lockdown. |
| | In addition, additional explanations have been added around the relevant sentences. |
| Line 444: please remove the "S" from "Saerosol". | Thank you for pointing it out. We have revised it. |
| Line 493: The statement regarding chemical alterations increasing the solubility of brake ring and tire wear debris requires supporting evidence. Which specific atmospheric processes or reactions contribute to such an increase in solubility? Please provide examples or references to substantiate this claim. | To the best of our knowledge, direct evidence for the chemical alteration of brake rings and similar materials in the atmosphere has rarely been reported. Nevertheless, in agreement with our findings, the PMF analysis of $PM_{2.5}$ sampled in Atlanta showed that approximately 35% of d-Fe was attributable to brake rings (Fang et al., 2015), and some reports have indicated that Fe in road dust (inclusive of brake rings) can be dissolved by |

sulfate from coal combustion (Wong et al., 2020). Furthermore, an unpublished graduation thesis suggests that Fe in brake pads demonstrates high solubility when exposed to acidic conditions (Simmons 2022). Consequently, Fe in brake pads has substantial potential for solubilization through atmospheric chemical reactions. The sentence structure of the relevant chapter has been largely modified, but the content is the same as that in the first version. Moreover, we have added further discussion as mentioned above. The revised text regarding the chemical alterations of tire rings is as follows:

As mentioned above, the aged dust fraction contained non-exhaust vehicle emissions (e.g., brake rings and tire wear debris), which were mainly present in the form of Fe-rich particles, such as Fe oxides (Sanderson et al., 2016; Li et al., 2022; Fu et al., 2023). Given that the $Fe_{sol}$% values of brake ring and tire wear debris were less than 0.01% in the absence of chemical alterations, including proton- and ligand-promoted dissolutions (Shupert et al., 2013; Halle et al., 2021), the increase in the [d-Fe]/[d-Al] ratio of the aged dust factor may have been caused by the dissolution of Fe from these materials during chemical alterations in the atmosphere. Previous research suggests that tire wear acts as an emission source of d-Fe in $PM_{2.5}$ (Fang et al., 2015), which can be dissolved by $SO_2$ emitted from coal combustion (Wong et al., 2020). Although further research is needed, our findings indicate that NEV particles, such as brake ring particles, can also be a source of d-Fe via aerosol acidification in the atmosphere.

| | |
|---|---|
| Section 3.7: The apparent discrepancy between the lack of a significant iron sulfate source in the PMF analysis and the observed dominance of Fe-sulfate particles in measurements is unclear. If these species play a major role in explaining D-Fe, why are their sources not represented as significant in the PMF results? What is the contribution of iron from the secondary aerosol factor? Could this factor be responsible for the Fe-sulfate signature observed? | We considered that Fe(II)-sulfate in fine aerosol particles mainly formed via aerosol acidification by $H_2SO_4$ rather than directly emitted from specific primary emission sources. Given that Fe(II)-sulfate forms through the alteration of aerosols involving various factors, like mineral particles and anthro-Fe, the contribution of Fe(II)-sulfate is unlikely to aggregate within a single specific factor. Regarding Fe in secondary sulfate aerosols, the primary emission sources of secondary sulfate aerosols were derived from $SO_2$ emitted from coal combustion. Therefore, Fe in the factor was likely emitted from coal combustion. As discussed in the main text, the d-Fe concentration in the secondary |

sulfate aerosol factor was higher than T-Fe likely due to the inclusion of d-Fe derived from mineral dust and steel industry factors. These findings are attributable to the correlation between d-Fe and sulfate concentrations. We have strengthened these points by adding the following discussion to multiple sections.

In the section of the "Monthly variation and size distributions of Fe species":

The water-soluble Fe species in fine aerosol particles likely formed through the chemical alterations of insoluble Fe, rather than directly emitted from primary sources. Although Fe(II, III)-sulfates are directly emitted from liquid fuel combustion, PMF analysis did not identify these emissions as the dominant source of Fe in fine aerosol particles. Similarly, Fe(III)-oxalate was not detected in the emission source samples of anthro-Fe. PMF results indicated that the total Fe in fine aerosol particles mainly originated from fresh and aged dust and the steel industry, with the dominant Fe species being primarily aluminosilicates and Fe-oxide nanoparticles. These Fe species were consistent with the insoluble Fe species identified in fine aerosol particles through XAFS spectroscopy. These primary sources, mineral dust, and steel industry–derived anthro-Fe typically exhibit low $Fe_{sol}$% without atmospheric chemical alterations. However, PMF analysis also revealed that aged dust and steel industry factors had a high $Fe_{sol}$%, highlighting the importance of the chemical alterations of Fe in mineral dust and anthro-Fe as key processes enhancing the water solubility of Fe in fine aerosol particles.

In the section of "Alteration processes and dissolution pH of mineral dust"

M1 spots exhibited low S intensity, and their Fe species were similar to those in mineral dust (aluminosilicates and Fe-(hydr)oxides; Figs. 9a and 9b). These findings were in accord with the μ-XAFS results for coarse aerosol particles. Furthermore, the SEM–EDX of aluminosilicates in coarse aerosol particles collected at the same observation point revealed low amounts of S (Sakata et al., 2021). As mentioned previously, PMF analyses indicated that fresh mineral dust was characterized by the

[nss-SO$_4^{2-}$]/[T-Fe] ratio of 0 (Tables S4 and S5). Therefore, the M1 spots in fine aerosol particles represented fresh mineral dust. By contrast, the XRF spectra of M2–M10 spots showed an intense S peak, and Fe-containing aluminosilicates were found to coexist with Fe(II, III)-sulfates, suggesting that these Fe-sulfates formed through the chemical alterations of Fe in mineral dust by H$_2$SO$_4$. This finding is supported by the PMF analysis, wherein the aged mineral dust factor included nss-SO$_4^{2-}$. Therefore, the internal mixing of Fe-bearing aluminosilicates with H$_2$SO$_4$ is a dominant process for the secondary formation of Fe-sulfates with high Fe$_{sol}$%. The average abundance of water-soluble Fe species (i.e., Fe(II, III)-sulfates and Fe(III)-oxalate) in M1–M10 was 46% ± 25%, which was higher than that in mineral-Fe$_{sol}$% (20.3%). This result is partly due to the small number of measurements of Fe species at points of low S intensity, such as the M1 spot.

In the section of "Alteration processes of anthro-Fe"

A previous study has demonstrated that Fe-rich particles (mainly Fe-oxides) collected directly from steel plants did not contain S but instead acquired a thick sulfate coating over one or two days of transport (Li et al., 2017). Given that the Fe-rich spots exhibited an intense S peak (Fig. 9a), these particles were markedly aged by SO$_2$ and/or H$_2$SO$_4$ in the atmosphere. Consequently, more than half of the Fe in these spots existed in the form of water-soluble Fe species, including Fe(II)-sulfate, Fe(III)-sulfate, and Fe(III)-oxalate (Fig. 9b).

Figure 10c: The panel is unclear. Does it suggest that there is no mineral-derived soluble Fe when pH > 2? If so, which dissolution pathways or processes are inhibited above this pH threshold? Clarification is needed.

The figure in the comment shows the pH dependence of Fe$_{sol}$% of mineral dust and anthro-Fe (i.e., mineral-Fe$_{sol}$% and antrrho-Fe$_{sol}$%). For the sake of clarity, the main text now specifies that Figure 10c presents the pH dependence of mineral-Fe$_{sol}$% and anthro-Fe$_{sol}$%, and I have expanded the discussion concerning these aspects.

The pH dependence of mineral-Fe$_{sol}$% and anthro-Fe$_{sol}$% is illustrated in Fig. 10c. Notably, when the pH was higher than 1.5, mineral-Fe$_{sol}$% was generally higher than anthro-Fe$_{sol}$%, whereas anthro-Fe$_{sol}$% exceeded mineral-Fe$_{sol}$% at pH levels lower

than 1.5 (Fig. 10c). This contrasting behavior occurred because the $Fe_{sol}$% of hematite nanoparticles, representing anthro-Fe, increased dramatically by approximately three orders of magnitude per unit decrease in pH (Eq. 12), leading to a surge in $Fe_{sol}$% from roughly 0.1% to 100% as the pH dropped from 2.2 to 1.2. By contrast, mineral-$Fe_{sol}$% gradually increased with decreasing pH (Fig. 10c). Consequently, anthro-$Fe_{sol}$% exceeded mineral-$Fe_{sol}$% within the pH range of 1.4–1.5 (Fig. 10c). In line with this pH-dependent behavior, the pH of samples wherein anthro-$Fe_{sol}$% exceeded mineral-$Fe_{sol}$% during the JPN period was considerably lower than 1.4 (average pH: 0.60), whereas that of the samples with high mineral-$Fe_{sol}$% during the EAout period exceeded 1.4 (average pH: 1.8). Therefore, given that mineral dust and anthro-Fe were emitted in the form of insoluble Fe, the relationship between mineral-$Fe_{sol}$% and anthro-$Fe_{sol}$% depended on the pH during reactions (Fig. 10c). This situation implied that the high $Fe_{sol}$% often seen in fine particles might not always be linked to anthro-Fe, making determining the origin of aerosol Fe solely on the basis of relative $Fe_{sol}$% levels difficult.

| | |
|---|---|
| Line 640: There is no Figure 11c. Did you mean to refer to Figure 10c instead? | I apologize for the inconvenience. I have corrected the figure numbers to the appropriate ones. |

---

## Author Comment (AC3)

Reviewer 1

| Reviewer's comments | Author's Reply |
|---|---|
| **Overall comments**: The manuscript presents a new concept discussing iron solubility of aerosol particles in the Pacific Ocean determined by atmospheric chemistry in East Asia. The work has been discussed in context to previous literature with appropriate references. However, the manuscript needs major revisions to improve clarity and structure for ease of interpretation. Key terminology has not been defined when introduced, with too many terms being used, causing confusion. Several figures have been incorrectly referenced and labeled throughout the publication. While the COVID-19 lockdown has been mentioned in the abstract and introduction, it lacks substantive discussion in the results and implications section, especially with respect to anthropogenic iron sources. | We sincerely thank the reviewer for the time and effort you have put into this review. We have carefully revised the manuscript with full consideration of the reviewer's comments and suggestions. Responses to the reviewer's comments are in black and not indented; corresponding revisions in the manuscript are in red and indented. |
| Line 104-108: Please add a clarification on when (pre or post sampling) and why filters were hydrophilized and treated with ethanol | During pre-sampling filter cleaning, the hydrophobic PTFE fiber filters did not sink in hydrochloric acid, making their proper cleaning difficult. Therefore, we hydrophilized the filters with ethanol so that they can be submerged in the acid. The ethanol evaporated during the air drying of the filters, returning them to a hydrophobic state at the time of sampling.

PTFE filters are not properly wetted by cleaning solutions because they are hydrophobic. This situation has the potential to reduce cleaning efficiency. Therefore, the filters were hydrophilized with ethanol (99.5%, Wako First Class, Wako, Japan). The hydrophilized PTFE filters were soaked in 1 mol L−1 hydrochloric acid (EL grade, Kanto Chemical Co. Inc., Japan) and heated at 180 °C for one day. Subsequently, the filters were placed in ultrapure water and heated at 180 °C for one day. The rinsed filters were then air-dried in a clean booth. Air drying restored the hydrophobicity of the PTFE filters as a result of the complete removal of ethanol from the filters. |
| Line 136: Explain what $(T\text{-}Fe/T\text{-}Al)_{aerosol}$ stands for in the equation? Clarify this term | We have added explanation of $(T\text{-}Fe)/(T\text{-}Al)_{aerosol}$.

The enrichment factor of Fe ($EF_{T\text{-}Fe}$) normalized by the mass ratio of Fe relative to that of Al in the upper continental crust (UCC) was calculated to evaluate the |

| | |
|---|---|
| | emission sources of Fe. The following equation was used for the calculation:

$$EF_{T-Fe} = \frac{(T-Fe/T-Al)_{aerosol}}{(T-Fe/T-Al)_{UCC}}, \quad (Eq.\ 1)$$

where (T-Fe/T-Al)aerosol represents the mass concentration of total Fe (= insoluble Fe + d-Fe in aerosol particles relative to the total Al). |
| Line 173: The y-axis should be correctly labeled as EF $_{T-Fe}$ to be consistent with the text | Thank you for your suggestion. We have revised the y-axis label. The revised figure was attached below.

 |
| Line 194: What are JPN+EAout, JPN, and EAout periods? Please specify the dates they comprise, as they have not been mentioned previously. | The sampling period in this study was separated into two categories: JPN, representing seasons dominated by air masses from within Japan, and EAout, representing seasons that are markedly influenced by air masses from East Asia, such as China. In the initial version of our manuscript, the detailed discussion on the origin of air masses was provided in Section 3.1. However, as pointed out in the comments, information on the origin of air masses was used in the PMF explanation. Therefore, Section 3.1 has been removed, and its contents have been moved to Section 2.1. |
| Line 261 Suggest using %Fe$_{max}$ instead of [%FeT] for clarity. Avoid using too many terms if possible, as it is confusing. | As per your suggestion, %Femax has been rephrased with [%FeT]. |

| | |
|---|---|
| Line 270: What is S/L ratio? | I apologize for not spelling out the S/L ratio. The S/L ratio refers to the solid-to-liquid ratio. The relevant sentence has been improved following.

Under the assumption that the solid-to-liquid ratio of anthro-Fe is 0.06 g L$^{-1}$, which is comparable to that of mineral dust, the aerosol liquid water (ALW) content associated with hematite nanoparticles was quantified by using the following equation. |
| Line 327: There is no need to indicate bar graph and line graph with axis; this is self-explanatory in the legend. | Thank you for your suggestions. We revised the figure as following your comment. The revised figure is attached below.

(a) d-Fe conc. and Fe$_{sol}$% of TSP |
| Line 350: Elaborate on what nss-SO42-/t-Fe represents before using the term. What is its importance? | I apologize for the insufficient explanation regarding the [nss-SO$_4^{2-}$]/[T-Fe] ratio. This ratio is used as an indicator of the acidity of iron-containing particles. An explanation of [nss-SO$_4^{2-}$]/[T-Fe] has been added to the relevant sentence.

Indeed, the Fe$_{sol}$% of coarse aerosol particles was correlated with the [nss-SO$_4^{2-}$]/[T-Fe] ratio as an indicator of the acidity of Fe-bearing particles (Fig. S5a; Zhu et al., 2020, 2022; Liu et al., 2022). |
| Line 351: Smallest particle diameter (< 0.39 μm) does not consistently seem to have higher solubility except maybe in Feb 2020, as opposed to what is stated. Please clarify this discrepancy | The relevant sentence describes that in coarse aerosol particles (>1.3 μm), Fe$_{sol}$% increased with the decrease in particle shape (increase in surface area). The following sentence has been revised to clarify that Fe$_{sol}$% is being compared within the coarse particle fraction. |

| | The $Fe_{sol}$% of the finest particles did not increase likely because this size fraction contained a large amount of fresh Fe-bearing particles that had not experienced acidic conditions. This result is supported by Fe speciation analysis using macroscopic XANES spectroscopy, which showed that Fe(III)-sulfate was not present in all samples. |
|---|---|
| Line 392: Clarify what figure is being discussed here (Presumably 6b)

Line 402: The figure caption for figure 6 is same as figure 5. Correctly describe Figure 6 and adjust the text accordingly.

Line 405: The yellow regions labeled as the JPN period in fig 5 and 6 are incorrect, which complicates interpreting the results. Fix the labels and discussion accordingly.

Line 630 and 640: Figures 10b and 10c have been incorrectly discussed as 11b and 11c throughout the text. There is no figure 11c. Please correctly state which figure is being referred to, and review the supporting text for consistency. | We sincerely apologize for the inconvenience caused by the errors in the figure numbers and captions. We have thoroughly reviewed the entire manuscript and corrected these issues. |
| Line 410: What is the chemical alteration being referred to? Is it only ocean acidification or other factors as well? Elaborate | We consider that aerosol acidification is the most dominant process to solubilize mineral dust in fine aerosol particles. Furthermore, an explanation for the limited impact of ligand-promoted Fe dissolution has been added at the end of the paragraph.

Notably, ligand-promoted dissolution is considered a process that increases the $Fe_{sol}$% of mineral dust. However, the contribution of ligand-promoted Fe dissolution was likely small because there were almost no plots of aerosol particles in region (ii) of Fig. 5b, a region where this process is a major contributor to mineral dust. |
| Line 427: Please add more discussion on Fesol% from anthropogenic sources during the COVID-19 lockdown period. The datapoint in Figure 6c for the lockdown period is missing. Does anthro-Fe% drop significantly during COVID-19 lockdowns, and what does this imply about the primary sources of soluble anthro-Fe (e.g., industrial vs. vehicular emissions)? | The $Fe_{sol}$% of anthropogenic Fe could not be calculated because the $EF_{T-Fe}$ and [d-Fe]/[d-Al] ratio of fine aerosol particles were lower than the representative values for mineral dust, indicating that anthro-Fe had no contribution to fine aerosol particles (computationally, the abundance of anthro-Fe became negative). Therefore, we have added the reasons for the missing plots of anthro-$Fe_{sol}$% during the COVID-19 lockdown for caption of Fig. 6c.

The plots of anthro-$Fe_{sol}$% in panel (c) are missing because either or both anthro-Fe or anthro-dFe concentrations were 0 due to the remarkable but small |

| | contributions of anthro-Fe during the COVID-19 lockdown period. |
|---|---|
| | Regarding to the primary source of anthro-Fe in fine aerosol particles, an important source of anthro-Fe, and its contribution factor during the COVID-19 period significantly decreased compared to other samples (Please see Figure S7). In the case of vehicular emissions, road dust encompassed in aged mineral dust factors (Figures 7b and 7e). Therefore, the factor contribution did not exhibit a reduction as pronounced as that of steelworks during the COVID-19 period. |
| Line 491: The term "Atmospheric Chemical alterations" is too vague. Specify the mechanisms driving Fe dissolution or discuss in more detail | Thank you for your comments. We have revised the relevant sentence in following.

 Given that the Fesol% values of brake ring and tire wear debris were less than 0.01% in the absence of chemical alterations, including proton- and ligand-promoted dissolutions (Shupert et al., 2013; Halle et al., 2021), the increase in the [d-Fe]/[d-Al] ratio of the aged dust factor may have been caused by the dissolution of Fe from these materials during chemical alterations in the atmosphere. |
| **Methods Section**: The methods are too lengthy and introduce many terms. Consider moving detailed protocols to the SI. | Some parts of the method section have been moved to Supplemental Information. |
| **Implications Section:** Please reword the implications section to place results in a broader context without introducing new terms like RDF that have not been mentioned previously. | In accordance with your feedback, the part concerning RDF has been deleted. The "Implication" section has been renamed to "Conclusions and future implications," and its content structure has undergone a substantial revision. Ther revised sentences were attached below.

 In this study, we investigated the factors controlling $Fe_{sol}$% in size-fractionated aerosol samples collected in the coastal region of the Sea of Japan. Our results showed that the T-Fe and d-Fe concentrations in TSP samples peaked in spring due to the substantial loading of Asian dust into the atmosphere. Steel industry and NEV particles, which were primarily composed of insoluble Fe, were important sources of T-Fe in fine aerosol particles. During the COVID-19 lockdown, the contribution of anthro-Fe (especially from the steel industry) to T-Fe decreased sharply, highlighting that anthro-Fe emitted from combustion and non-combustion sources was a major source of T-Fe in fine aerosol particles over East Asia. $Fe_{sol}$% was |

higher in summer than spring, with high values mainly observed in fine aerosol particles, and correlated with the [nss-$SO_4^{2-}$]/[T-Fe] ratio, indicating that Fe in these fine particles was primarily dissolved through proton-promoted dissolution. Macroscopic and microscopic XANES spectroscopy revealed that the water-soluble Fe species in fine aerosol particles were Fe(II)-sulfate, Fe(III)-sulfate, and Fe(III)-oxalate and were also present in mineral dust and anthropogenic aerosols. Given the water insolubility of Fe species in freshly emitted mineral dust (aluminosilicates) and anthro-Fe (mainly Fe oxides), these water-soluble Fe species likely formed through aerosol acidification by $H_2SO_4$, a process supported by the strongly acidic conditions suggested by dissolution pH estimations. Therefore, chemical reactions, including aerosol acidification, play a critical role in the control of the $Fe_{sol}$% of aerosol particles in East Asia.

During the period of increased aerosol outflow from East Asia (November to April), the average $Fe_{sol}$% of TSPs collected at NOTOGRO (4.9%) was slightly lower than that of TSPs collected in the North Pacific. However, the $Fe_{sol}$% of fine aerosol particles increased substantially during transportation from East Asia to NOTOGRO, with their average $Fe_{sol}$% (14.3%) being comparable to that of fine aerosol particles collected in the western Pacific during a similar season (14.2%; Table S3). This finding suggests that the chemical alterations of Fe in mineral dust and anthro-Fe in fine aerosol particles mainly occurred over East Asia rather than during transport in the North Pacific. Therefore, long-term observations on the $Fe_{sol}$% of the fine aerosol particles collected at the rim of East Asia (i.e., entrance of the North Pacific) play an important role in understanding the controls on $Fe_{sol}$% supplied to the North Pacific. By contrast, the $Fe_{sol}$% of coarse aerosol particles were slightly higher in the western Pacific (average: 3.5%) than in NOTOGRO (average: 0.5%). This difference likely contributed to the difference in the $Fe_{sol}$% of TSPs between the two regions. Therefore, future research should also focus on the Fe dissolution processes in coarse aerosol particles during transport over the marine atmosphere to develop our understanding of aerosol Fe supply to the ocean

| | surface because these differences may be a reason for the higher $Fe_{sol}$% of TSPs in the western Pacific than in East Asia. |
|---|---|
| **Minor comments:**

Line 34: Use another word instead of 'outside' (e.g., except).

Line 73, 89: Subscript 'sol' in $Fe_{sol}$%

Line 596: Correct the spelling of 'Dissolution' and 'Mineral' in Figures 10a and 10b. | Thank you for your checking. These issues have been improved. |